# Hydrostatic Water Displacement Sensing for Continuous Biogas Monitoring

**DOI:** 10.3390/s25237297

**Published:** 2025-11-30

**Authors:** Marek Habara, Jozef Molitoris, Barbora Jankovičová, Jan Rybář, Ján Vachálek

**Affiliations:** 1Faculty of Mechanical Engineering, Slovak University of Technology in Bratislava, Námestie Slobody 17, 812 31 Bratislava, Slovakia; xmolitoris@stuba.sk (J.M.); jan.rybar@stuba.sk (J.R.); jan.vachalek@stuba.sk (J.V.); 2Faculty of Chemical and Food Technology, Slovak University of Technology in Bratislava, Radlinského 9, 812 37 Bratislava, Slovakia; barbora.jankovicova@stuba.sk

**Keywords:** anaerobic digestion, biogas, cloud data acquisition, continuous monitoring, hydrostatic water displacement, IoT platform, PID temperature control, pressure measurement

## Abstract

Biogas and biomethane represent promising domestic fuels compatible with decarbonization targets at a time when diversification of gas sources is essential due to market volatility and increasing security risks. In laboratory practice, however, biogas production is still frequently assessed manually, which increases measurement uncertainty, limits temporal resolution, and reduces comparability between experimental series. We present an open and low-cost platform for continuous monitoring based on the hydrostatic water-displacement principle, complemented by stabilized process conditions (temperature control at 37 °C with short-term variability of approximately ±0.02 °C), continuous measurement with a 1 Hz sampling rate, and cloud-based data visualization. The methodology builds on a standardized procedure grounded in well-defined pressure–height–volume conversion relationships and transparent signal processing, enabling objective comparison of substrates and experimental setups. Validation experiments confirmed the system’s capability to capture short-term transient phenomena, improve reproducibility among parallel reactors, and maintain long-term measurement stability. Long-duration tests demonstrated short-term scatter of approximately 0.06 mL, minimal drift below 0.15% per 24 h, and an expanded uncertainty of roughly 3.1% at 100 mL. In parallel BMP tests, the continuous method yielded final volumes 5.78% higher than the discrete pressure method, reflecting systematic bias introduced by sparse manual sampling and reactor handling. The basic configuration quantifies the cumulative volume and production rate of biogas and is readily extendable to online gas composition analysis. The proposed solution offers a replicable tool for research and education, reduces costs, supports measurement standardization, and accelerates the optimization and subsequent scale-up of biogas technologies toward pilot-scale and industrial applications.

## 1. Introduction

Growing instability in energy markets and recurrent disruptions in supply chains highlight the need to diversify gas sources and strengthen energy security. In addition to imports of fossil natural gas, increasing attention is being directed toward domestic, low-carbon, and flexible fuels that are compatible with existing infrastructure and contribute to decarbonization goals. In this context, biogas and biomethane represent promising options: they are storable, dispatchable, can be upgraded to grid-injection quality, and, when properly managed, can reduce net methane emissions. They also support the principles of the circular economy, making them an essential component in the diversification of gaseous fuel portfolios with respect to system resilience, regional self-sufficiency, and price stability [1].

The successful development of biogas and biomethane production, however, depends on reliable and standardized monitoring of anaerobic digestion (AD) processes. AD performance is highly sensitive to substrate characteristics, inoculum composition, temperature regimes, and inhibitory factors. Traditional evaluation methods, such as manometric and volumetric approaches, rely on discontinuous measurements that provide limited temporal resolution. Manual readings are typically taken only every few hours or days, preventing the observation of dynamic changes and the timely detection of short-term transient events. Moreover, these methods are prone to operator-induced errors. Quantitative analyses have shown that manual manometric measurements of biomethane potential (BMP) may exhibit a negative bias of up to 24% in reactors with a small headspace volume [1]. Additional losses of 25–30% of biogas have been reported due to leakage of laboratory septa when pressures exceed 1 bar [1]. These limitations reduce comparability between laboratories. Inter-laboratory studies have repeatedly demonstrated low reproducibility, with BMP values for the same substrate differing by more than 100% [2]. Such variability slows down process optimization and undermines the reliable transition from laboratory testing to pilot and industrial applications.

To overcome these limitations, several approaches have been proposed. At one end of the spectrum are commercial high-precision measurement systems based on digital thermal mass flow controllers, such as the Bronkhorst EL-FLOW Select series [3]. These devices provide excellent accuracy, stability, and fast response times, but their high purchase cost and need for professional maintenance are limiting factors for many laboratories. At the opposite end are semi-automated and low-cost systems that seek to reduce human error and ensure continuous data acquisition. Examples include pressure-based monitoring devices built around microcontrollers (e.g., Arduino or ESP32), designed for biogas pressure tracking in batch digestion tests [4]. A similar approach was presented by Pérez-Vidal et al., who demonstrated the feasibility of continuous manometric monitoring using a microcontroller; however, their implementation still relied on accumulated headspace pressure.

Consequently, there is a growing demand for robust, accurate, and cost-effective monitoring systems that can capture dynamic process behavior, provide reproducible data, and offer an economically viable alternative to expensive commercial equipment. Modern approaches in environmental engineering increasingly focus on the automation of measurement procedures, reduction in operator intervention, and open-source platforms that support reproducibility and research transparency [4]. A comparison of the limitations of traditional methods versus the advantages of automated approaches is presented in Table 1.

In this study, we present a modular system for continuous monitoring of biogas production. We describe its operating principle, validate its performance in long-term experiments, and compare its results with traditional methods. The aim is to demonstrate that automated monitoring based on an open and low-cost design can enhance measurement accuracy and comparability, accelerate experimental validation, and provide a reliable tool for advancing biogas technologies toward pilot-scale and industrial applications.

## 2. Materials and Methods

In laboratory conditions, biogas production is traditionally evaluated using two manual approaches—the volumetric and the pressure method (Figure 1). The volumetric method is among the simplest techniques, based on direct observation of the gas volume that displaces water from a measuring device, typically a burette. The reactor is sealed with a rubber stopper containing a needle connected to a water-filled burette; the generated gas creates overpressure and gradually displaces the liquid, while the gas volume is determined manually by reading the level displacement after stabilization [5]. The pressure method, on the other hand, is based on gas accumulation in a closed reactor and subsequent gas transfer via a gas-tight SGE syringe into a water-filled burette. The overpressure in the syringe pushes the liquid upward, and the displaced water volume corresponds to the amount of accumulated gas [6].

Although these methods are widely used for their simplicity and low technical demands, they introduce several limitations. As discontinuous measurements, they provide only sporadic data, incapable of capturing short-term process dynamics or transient phenomena during anaerobic digestion. The requirement for manual intervention increases the risk of subjective errors and sensitivity to operating conditions—particularly temperature fluctuations and partial vapor pressures. These factors significantly reduce the reproducibility and comparability of results between series and laboratories, slowing down process optimization and the transfer of knowledge to pilot and industrial applications [7,8].

### 2.1. Concept of Automated Biogas Production Measurement

The concept of automated biogas production monitoring arises from the need to overcome the limitations of traditional discontinuous approaches and to ensure continuous, accurate, and reproducible data. The primary objective is to develop a system capable of maintaining stable fermentation conditions over long periods, continuously measuring the produced gas, and reliably processing and storing the recorded data. Stabilization of process conditions is essential, as anaerobic digestion is sensitive to temperature deviations and uneven substrate mixing. Therefore, the concept includes an enclosed reactor chamber equipped with active heating and feedback control to maintain the mesophilic regime with an accuracy of ±0.5 °C. Additional features include basic thermal insulation and gentle mixing to prevent sedimentation of solids and maintain homogeneous conditions throughout the reactor volume.

The second requirement is to ensure continuous measurement of gas production. The concept is based on the hydrostatic principle, in which the generated gas displaces the liquid in a communicating U-tube, and the resulting level shift is converted into a pressure difference measured by a pressure sensor (Figure 2). The microcontroller samples the pressure sensor at a higher internal frequency, while the proposed data-logging frequency for cloud upload is 1 Hz. A higher upload rate is not meaningful, as gas production dynamics in laboratory anaerobic digestion change on a scale of minutes rather than milliseconds, meaning that even 1 Hz far exceeds the temporal resolution required for process interpretation (additional technical details regarding sampling and data handling are provided in Section 2.3). This enables continuous recording of gas production with sufficient sampling frequency to capture process dynamics without the need for manual intervention. To maintain consistent geometry and eliminate drift caused by evaporation or level shifts, the concept includes cyclic venting of the accumulated gas and automatic liquid replenishment. The result is a reliable and reproducible production record that allows monitoring not only of total volumes but also of process dynamics, including short-term transients.

The third component of the concept focuses on data processing and management. Measurements are continuously evaluated and stored on a cloud platform that provides real-time visualization, remote access, and data export for further analysis. This design eliminates subjective errors associated with manual readings and creates a data foundation suitable for statistical evaluation, series comparison, and long-term trend monitoring. The system is designed from the outset to be modular and replicable: each reactor is equipped with an independent measuring branch, allowing simultaneous testing of multiple substrates under identical conditions. The basic configuration monitors gas volume and production rate, while the architecture allows future expansion with sensors for online gas composition analysis. In this way, the system forms a universal platform that can serve not only research and education but also as a foundation for standardization and scaling of biogas technologies from laboratory to pilot and industrial levels.

### 2.2. Mechanical and Fluidic Architecture of the System

The proposed system is designed as a low-pressure fluidic loop in which the produced biogas flows directly from the reactor vessel into the measuring section shaped as a U-tube (Figure 3). One end of this tube is connected to the gas line, while the other end is open to the atmosphere, with a water column between the two arms. The generated gas displaces the water downward in the connected arm, causing the water level on the opposite side to rise proportionally. The change in water level is converted into a pressure difference, which is then interpreted as the volume of produced gas. The measuring branch is constructed from hoses with an inner diameter of 8 mm, significantly reducing the exposed water surface area and thus minimizing evaporation. In the proposed geometry, only one arm of the U-tube is open to the atmosphere, and the water–air interface corresponds to a circular area of approximately 0.5 cm^2^ (≈5 × 10^−5^) m^2^. Typical evaporation rates for quiescent water surfaces in laboratory environments are on the order of a few milliliters per square meter and hour (≈5 mL·m^−2^·h^−1^), a value empirically observed in our laboratory under standard indoor conditions. When applied to the exposed area of the U-tube, this corresponds to an evaporative loss of about 2.5 × 10^−4^ mL·h^−1^, i.e., less than 0.01 mL per day. This is several orders of magnitude lower than the 5 mL measurement increment, confirming that evaporation is negligible in the context of long-term measurement stability. The measuring geometry is fixed, protected by a plexiglass cover, and fully enclosed during operation, eliminating the need for operator intervention after initial filling and degassing of the system.

Operation is organized into repeating cycles consisting of three steps:Gas accumulation and rise in the liquid level;Automatic venting of gas through an electromagnetic valve;Level inspection and correction.

Each cycle is defined by a fixed gas volume of 5 mL. This approach has two major advantages—it keeps the measurement within the linear range of the sensor and minimizes the effect of evaporation, which during a single 5 mL step is negligible (under normal conditions, evaporation accounts for only a fraction of a milliliter per day). During long-term operation, a few milliliters of water loss may occur; this is automatically compensated by the peristaltic pump, which refills a defined amount of distilled water to maintain a constant reference level. This method eliminates the risk of systematic errors and ensures data consistency during multi-day experiments.

The entire fluidic architecture is designed with safety and ease of maintenance in mind: the gas lines have minimal dead volume, the system operates within a low overpressure range, and vented gas is discharged into a safe, well-ventilated area.

The structure of the following methods sections reflects the layered architecture of the system. Section 2.2 describes the fluidic arrangement and measuring geometry, Section 2.3 outlines the sensing and control electronics, and Section 2.4 presents the mechanical integration of all components. Together, these sections provide a coherent and complete description of the device, with each part addressing a distinct subsystem within the overall design.

### 2.3. Sensors and Control Electronics Used

The measuring subsystem is based on pressure sensors of the ABP2 (Honeywell) series, which communicate over a digital I^2^C interface with an internal 24-bit ADC. The selected ±60 mbar differential model provides an effective usable resolution of 14–16 bits, corresponding to a nominal sensitivity of about 0.01 mbar and a detectable hydrostatic level change of ~0.1 mm [9]. In the U-tube geometry with an 8 mm inner diameter, this translates to a theoretical volumetric sensitivity on the order of 0.05–0.1 mL. Such fine resolution enables reliable monitoring of even low daily gas productions, while the actual operation is organized into 5 mL increments to keep the signal in the linear range and to suppress the influence of evaporation.

According to the manufacturer’s datasheet, the ABP2 series are piezoresistive silicon pressure sensors with an on-board ASIC providing full factory calibration and temperature compensation for offset, span, non-linearity, repeatability and hysteresis. Calibrated values are internally updated at approximately 200 Hz [9]. In our implementation, the sensor is read using the reference I^2^C routine supplied with the Honeywell application note, resulting in an effective microcontroller sampling frequency of ~11 Hz per channel. Processed data are logged and transmitted to the cloud at 1 Hz, which already exceeds the temporal requirements of batch anaerobic digestion, where relevant dynamics occur on the scale of minutes to hours. Before filling the U-tube with water, a zero-offset check was performed to confirm correct baseline calibration; no user-level recalibration was required beyond this step, as the system relies on the factory-calibrated characteristics of the sensor.

All pressure sensors have fixed I^2^C addresses and are therefore connected through an I^2^C multiplexer, which ensures independent readings from four parallel reactors without transmission interference.

In addition to pressure sensing, two NTC temperature sensors (KY-013 modules, each containing a 10 kΩ B ≈ 3950 thermistor) are installed inside the reactor chamber to monitor the thermal environment. Although their nominal accuracy is specified as ±0.5 °C, low-cost NTC sensors typically exhibit unit-to-unit variability [10]. To quantify this, both sensors were examined in a dry-chamber calibration setup together with a laboratory thermometer at mesophilic temperatures 35–40 °C. The two modules showed small but distinct offsets: one deviated by approximately +0.11 °C, while the second exhibited an offset of about +0.18 °C. No appreciable drift was observed during the equilibration period, and both offsets were compensated individually in software using fixed correction factors. The sensors are mounted symmetrically on the rear wall to detect potential thermal gradients and to support monitoring of temperature homogeneity during anaerobic digestion.

The system is controlled by an Arduino Nano 33 IoT, which integrates a 32-bit ARM Cortex-M0+ SAMD21 microcontroller (48 MHz) together with the u-blox NINA-W102 Wi-Fi/IoT module. This platform was selected because it provides stable I^2^C communication for multi-sensor acquisition, low electrical noise on the digital supply rails, and native cloud-connected functionality required for autonomous long-term operation. Through its IoT connectivity, the controller ensures reliable data upload, remote access and system supervision, while simultaneously managing all local actuation tasks. Its processing capacity is fully sufficient to maintain deterministic timing across four parallel measurement branches [11].

The solenoid valves (3/2 type, SMC) operate at 12 V and allow easy switching between the measuring and venting phases [12]. The peristaltic pumps and magnetic stirrers are also powered from the 12 V branch, with their control handled via relay modules powered at 5 V [13,14,15]. A DC/DC step-down converter (12 V → 5 V) is used to relieve the microcontroller from directly supplying multiple relays [16]. The heating elements located in the reactor chamber are rated to maintain stable mesophilic temperatures at full volume and are switched using MOSFET transistors. This configuration allows safe control of higher currents through a low-level logic signal from the microcontroller [17,18]. The entire system is powered by a 360 W power supply, providing sufficient reserve for the simultaneous operation of all valves, pumps, stirrers, and heating elements [19]. In practical terms, the system operates at roughly 75% of the PSU’s rated capacity, leaving a ~25% performance margin to safely absorb inrush currents, valve-switching spikes, and long-term thermal loads. The architecture thus combines sensitive process sensing with robust control, forming the foundation for long-term, stable, and replicable operation (Figure 4).

### 2.4. Mechanical Design of the Device

The presented device (Figure 5) is precisely designed for accurate monitoring of biogas production from experimental reactors. Its robust structure is primarily composed of modular aluminum T-profiles forming a dual-chamber system: an upper incubation chamber and a lower measuring chamber. For easy transport and handling, four handles are integrated, strategically positioned—two on each side of the device.

The upper chamber serves as a closed, thermally insulated incubation chamber (Figure 6). Its walls are constructed as sandwich panels with inner and outer layers of technical plastic enclosing an insulating styrodur core, effectively minimizing heat losses and maintaining stable internal temperatures. Access to this chamber for reactor placement is provided by transparent, double-hinged plexiglass doors mounted on the front panel. Inside the chamber, heating elements with integrated fans are strategically mounted on the upper walls to ensure rapid and uniform air heating. Each reactor is securely positioned in a custom-designed holder with numerical labeling for easy identification. Below each holder, a DC motor with a flange containing two permanent magnets drives a magnetic stirrer inside the reactor, ensuring thorough mixing of the reactor contents. For precise temperature monitoring and assessment of thermal homogeneity, two NTC temperature sensors are symmetrically mounted on the rear wall of the upper chamber. On the outer top surface of the chamber, four quick coupling connectors (for hoses with an inner diameter of 4 mm) are integrated to serve as outlets for vented gas from the 3/2-way valves, allowing connection to either a digester or gas collection bags according to experimental requirements.

Behind the upper chamber, four 3/2-way solenoid valves (SMC VT307-6D1-02F-Q, SMC Corporation, Tokyo, Japan) are installed, each dedicated to a single reactor. Each valve is connected to three silicone hoses with an inner diameter (ID) of 4 mm: one inlet hose from the reactor (typically connected via a syringe needle fitting), one vent hose leading to a quick coupling connector on the roof of the upper chamber, and one measuring hose directing the biogas to the measuring system in the lower chamber.

The lower chamber contains the sensitive system for biogas measurement (Figure 6). Its rear wall and bottom are made of opaque technical plastic, while the remaining three sides are transparent plexiglass. The front plexiglass panel, positioned parallel to the doors of the upper chamber, is designed to be removable for maintenance purposes. Inside, cable ducts are used to ensure organized and protected routing of electrical wiring for various sensors and pumps.

The core of the measuring system in the lower chamber is the U-tube, which plays a key role in quantifying biogas production. This U-tube is custom-built from flexible hoses with an inner diameter (ID) of 8 mm (SMC TU1208BU-20, SMC Corporation, Tokyo, Japan), connected using L- and T-type quick fittings, with strict vertical alignment maintained by specially designed parallel holders. Biogas from the 3/2-way valve is conveyed through an ID 4 mm hose into the lower chamber, where it connects to a reduction quick fitting (ID 4 mm to ID 8 mm). From this reduction, it continues vertically downward through an ID 8 mm hose forming the first arm of the U-tube. It then connects to an L-fitting, followed by a short horizontal segment leading to a T-fitting. From this T-fitting, one vertical port continues upward via an ID 8 mm hose, forming the second arm of the U-tube. The horizontal port of the same T-fitting provides a connection point for the differential pressure sensor. The second arm of the U-tube is fixed at the top with another T-fitting. The upper vertical port of this T-fitting is intentionally left open to the atmosphere, serving as the reference pressure port. This open design is critically important because it ensures that only the gravitational force acts on the rising water column, preventing the formation of counterpressure and guaranteeing accurate measurement of the hydrostatic pressure. The lateral (horizontal) port of this upper T-fitting is connected, via reduction fittings, to the outlet of the peristaltic pump.

The system includes four individual peristaltic pumps (Grothen G328, Shenzhen Qiyu Technology Co., Ltd., Shenzhen, China; suitable for ID 4 mm tubing), each dedicated to one measurement channel. These pumps are used to replenish distilled water into their respective U-tubes as needed, compensating for any evaporation losses that may occur during extended experimental periods. Although the pumps are hydraulically connected via a common supply line (ID 4 mm hose), their inherent peristaltic pumping principle acts as a check valve, preventing backflow or siphoning between channels. This shared supply line terminates at a panel-mounted quick coupling connector on the left side of the lower chamber, facilitating connection to an external distilled water reservoir. On the same left wall, a control panel with four buttons allows manual operation of each peristaltic pump. This manual mode is essential for the initial filling of the U-tubes and for critical degassing (removal of trapped air from the small tube leading to the differential pressure sensor) before starting the experiments.

A differential pressure sensor is used for precise measurement of the hydrostatic pressure of the water column. The second (reference) port of this sensor is not left open but connected to a short damping tube. This short tube effectively attenuates rapid fluctuations in ambient pressure, contributing to more stable and reliable pressure readings. All four differential pressure sensors are connected to an I^2^C expander, which is also located in the lower chamber, simplifying data acquisition. On the outer rear wall of the lower chamber, a DC power supply and a distribution box containing all control electronics are mounted, with the programming connector conveniently routed to the front panel of the distribution box for easy access. The entire mechanical design was created using CAD software SolidWorks 2022.

### 2.5. Control Strategy and Software Architecture

The device is controlled by four independent state machines (one per reactor), operated through a common sampling loop (Figure 7). The I^2^C multiplexer addresses the pressure branch channel, the raw signal is read, and the corresponding state machine is evaluated immediately. The measurement cycle consists of four steps:Accumulation—the generated gas increases the hydrostatic pressure in the U-tube, and the system continuously integrates the volume;Venting to the atmosphere—brief opening of the 3/2 valve equalizes pressure conditions (fixed time window);Refilling—the peristaltic pump restores the distilled water level to the reference point (activated only after triple confirmation of low level from consecutive samples);Stabilization—a short time window to suppress transient effects after valve/pump actions.

Transition to venting is triggered when a fixed volumetric step of 5 mL is reached, keeping the measurement within the linear range of the sensor and reducing the effect of evaporation within a single step to a negligible level. Sampling is time-separated from actuator operations to minimize artifacts. For each reactor, the cumulative volume is maintained, and once every 24 h the daily increment (difference between consecutive cumulative values) is calculated. A branch reset to a known state with brief initial venting and a manual mode for safe refilling with overflow protection are also available.

Control of the reactor chamber conditions runs in parallel and is synchronized with the measurement cycle to avoid interference during critical sensing moments. Temperature is measured by two independent sensors and regulated using a PID algorithm with a computation period of approximately 1 s. The deviation between the target and actual temperature is converted into PWM for the heating branches: the proportional term covers the instantaneous error, the integral term increases under positive deviation and is actively relieved under negative deviation (anti-windup), while the derivative term suppresses rapid fluctuations [20]. This ensures stability of the mesophilic regime within approximately ±0.5 °C without significant overshoot. Substrate mixing occurs continuously during monitoring at low speed to promote homogeneity and gas release without introducing vibrations into the measuring branch. Start, reset, and manual mode switching are available for each reactor; status indicators signal activities such as refilling or active monitoring and allow retrospective pairing of interventions with responses in the data.

Acquisition and telemetry are designed to preserve metrological integrity. The device transmits raw time series to the cloud platform—derived gas volume from pressure, temperature, actuator states (valve, pump, mixing, heating), and daily increments. No local filtering of measured data is applied to prevent double signal processing; thus, data analysis belongs to the analytical part of the work. Practical protections integrated into the state machine include a time delay before starting monitoring after valve/pump actions (to suppress transient effects), a short valve reset after branch initialization, triple verification of low-level detection before refilling, and saturation of negative production increments to zero (to prevent non-physical entries caused by noise).

To ensure continuity of data collection, the current prototype streams raw time series directly to the cloud platform in real time. In the present version, no local buffering is implemented; therefore, temporary loss of connectivity results in a gap in the transmitted dataset. The software architecture, however, already includes reserved interfaces for storage expansion, and the next design iteration will integrate an onboard SD-card logger to provide local redundancy and guarantee uninterrupted recording even during extended network outages.

Regarding operational safety, the control logic currently incorporates only fundamental software protections (e.g., rejection of negative increments, minimum stabilization times, triple-verified level detection, and initialization venting). Hardware-level failures such as valve malfunction, excessive pressure increase, or unexpected absence of water replenishment are not monitored explicitly. These features are planned for the next system generation, where overpressure detection using the existing sensor range, valve-state consistency checks, and notification routines will be incorporated into the supervisory layer of the control logic.

### 2.6. Signal Processing and Measurement Uncertainties

The signal from the pressure sensors is transmitted in digital form via the I^2^C interface, eliminating the need for additional A/D conversion and minimizing the influence of electrical noise. Each reading represents the current pressure difference between the arms of the U-tube. The sensor includes integrated temperature compensation of offset and sensitivity (through an ASIC), so the effect of temperature on the measured pressure value is largely eliminated at the hardware level. Therefore, no additional numerical correction of water density is required during signal processing.

The measurement chain is based on converting pressure into the height of the liquid column and subsequently into the volume of water displaced in the U-tube. In the first step, the height of the equivalent water column is calculated from the fundamental hydrostatic relation:(1)$$h=\frac{\Delta p}{\rho \;\cdot \;g}$$
where h is the height of the water column (mm), ∆p is the measured pressure difference (Pa), ρ is the density of water (kg·m^−3^), and g is the gravitational acceleration (m·s^−2^) [21].

The height of the water column is then transformed into the volume of displaced water in the U-tube arm:*V = A·h*(2)
where V is the volume (mL), h is the height (mm), and A is the internal cross-sectional area of the hose calculated from the known diameter (8 mm) [22].

No local filtering or smoothing is applied to the data. All measured values (pressure, volume, temperature, actuator states) are stored in raw form in the cloud database, preserving the full metrological integrity of the data. Outlier removal, consideration of influencing parameters, and statistical analysis are performed only in post-processing to avoid double handling of the data.

Determining the overall accuracy of the measurement system requires a comprehensive estimation of measurement uncertainty. This uncertainty arises from a combination of several independent components, which include not only the physical characteristics of the system (pressure sensor accuracy, physical properties of the media, geometric tolerances) but also factors related to process dynamics and data processing methodology. In accordance with international standards, the combined standard uncertainty ($u_{c}$) is expressed as the root-sum-of-squares of the individual uncertainty components:(3)$$u_{c}=\sqrt{\sum_{i=1}^{N}{u_{i}}^{2}}$$

This relationship summarizes all identified and quantified sources of uncertainty [23]. A detailed metrological analysis, the complete uncertainty budget, and the quantification of individual components for the developed platform are presented separately in Section 3.5.

The conversion from bar to pascal is applied according to the relation:1 bar ≈ 100,000 Pa

This allows all quantities and their units to be expressed uniformly in the SI system and directly applied in calculations [24]. Based on the system geometry (hose diameter 8 mm), it can be derived that the minimum detectable pressure changes correspond to increments of approximately tenths of a milliliter of gas, which is negligible in the context of typical daily productions on the order of tens to hundreds of milliliters.

The processed data thus represent the input for further analyses. Unlike traditional manual methods, the comparison does not use raw data directly but rather data processed with the inclusion of estimated influencing parameters from the conducted experiments. This approach ensures metrological correctness and enables an objective evaluation of differences between the automated platform, manual methods, and model predictions.

### 2.7. Experimentálny Protokol

The experiments were designed to verify the functionality of the automated system while simultaneously providing comparable results with traditional monitoring methods. Glucose (C_6_H_12_O_6_), a simple monosaccharide readily biodegradable under anaerobic conditions, was selected as the substrate. The inoculum consisted of anaerobically stabilized sludge from a wastewater treatment plant (VS 13 g/L, vs. = volatile solids), assumed to have an optimal content of nitrogen and phosphorus necessary for microbial activity. The total reactor volume was 310 mL, and the dosed mixture consisted of 160 mL of anaerobic sludge and 1 g of glucose as the substrate. The prepared mixtures were distributed into four parallel reactors, with three containing the same substrate and the fourth serving as a blank without substrate addition. These four reactors therefore represented biological replicates operated under identical thermal, mixing, and measurement conditions. Repeatability was later assessed by comparing cumulative production curves and daily increments among replicates.

Initial conditions included homogeneous mixing of the inoculum and substrate and setting of mesophilic conditions (37 °C). Mixing was performed mechanically at a constant speed of 14 rpm to prevent sedimentation of solid particles and to promote the release of gas bubbles. The system allows the produced gas to be discharged directly into the digester, while alternatively, it can be collected in gas bags for subsequent laboratory composition analysis.

Data collection was performed continuously throughout the experiment. Pressure-derived gas volume was sampled internally at approximately 11 Hz and logged to the cloud platform at 1 Hz. This logging frequency is substantially above the rate of gas formation dynamics and therefore provides oversampling with no loss of temporal resolution.

Although laboratory digestion tests typically last 7–10 days or longer depending on substrate degradability, the present validation experiment was intentionally limited to 90 h. This duration was selected to enable a controlled comparison between the automated system and the reference method under stable mesophilic conditions. The end of each run was determined from the shape of the cumulative gas production curve, and the experiment was concluded once the production rate stabilized at minimal values.

As part of the protocol, the same samples were tested in parallel using the traditional pressure-based water-displacement method, which follows the hydrostatic principle described in standard anaerobic digestion procedures (e.g., VDI 4630 [25]). This provided a direct reference for assessing accuracy and practical agreement between manual and automated readings. The results from both sources—the automated measurement and the manual methods—were then compared to evaluate the accuracy, reproducibility, and practical added value of the new platform.

### 2.8. Data Acquisition and Monitored Variables

The proposed system allows monitoring of up to four reactors simultaneously, with the user able to select how many of them will be active in a given experiment. Each reactor has its own dedicated measuring branch, and all active branches are logged independently. The primary measured variable is the hydrostatic pressure of water in the U-tube, which is converted to the volume of produced gas. Each data point is time-stamped, creating a continuous time series for each reactor. Data collection and visualization are handled via the Arduino Cloud platform with remote monitoring capability; the data can be exported in a universal CSV format for subsequent analysis (e.g., in Python, MATLAB, or spreadsheet software).

Data recording occurs at two temporal resolutions. Internal sampling of the pressure sensor is performed at approximately 11 Hz, while data are transmitted and stored on the cloud platform at 1 Hz, corresponding to 60 samples per minute (3600 samples per hour) for each monitored variable. This sampling density substantially exceeds the dynamics of gas formation during mesophilic digestion and therefore ensures oversampling without loss of temporal detail.

The cumulative gas production volume is continuously updated and appended to the dataset. Simultaneously, every 24 h, the daily increment is calculated as the difference between two consecutive cumulative values. This approach provides a quick indicator of daily production without the need for additional time-series processing.

As supplementary variables, temperatures from two sensors placed in the reactor chamber are also recorded. These serve as a control mechanism and are used only when deviations appear in the production data that could be related to thermal effects. The states of actuators (valves, pumps, stirrers, heating elements) are not stored in the dataset; they are intended solely for operator visualization during operation.

The resulting data files therefore contain timestamps, cumulative gas volume, 24 h increments, and temperatures from all active reactors. This data structure ensures unambiguous interpretation, high reproducibility, and direct usability for comparison with traditional monitoring methods.

In the current prototype version, no automatic alerting or data integrity notification is implemented in the cloud dashboard. Real-time visualization is available at all times, but loss of connectivity results in a temporary interruption of data transmission. Since the primary objective of this prototype was to validate the measurement principle and long-term operational stability, no additional redundancy layer was implemented at this stage. The modular software architecture, however, allows these supervisory functions to be added in future system iterations.

## 3. Results

This section summarizes the results of the experimental validation of the proposed platform for biogas production monitoring. The objective is to present the extent to which the proposed solution meets the requirements for reliability, accuracy, and practical applicability under laboratory conditions. The first part therefore focuses on verification of the device’s correct operation, emphasizing the validation of all key modules as well as the assessment of operational stability and measurement consistency over time. This section also includes a demonstration of remote cloud visualization capabilities, which represent an important feature for long-term experiment monitoring.

The initial verification analysis is followed by the presentation of representative datasets obtained from the automated measurement system during controlled experiments. These data provide a detailed view of process dynamics and form the basis for critical comparison with the results of manual reference methods. A separate subsection elaborates on the metrological aspects, which constitute the essential framework for accuracy evaluation. Parameters influencing the measurement results and the issue of reproducibility in repeated experiments are discussed.

The final part of this section is structured as a comparative analysis of the automated and manual approaches. The analysis focuses on the strengths and weaknesses of both methods, assessing not only their accuracy but also their practical usability, time efficiency, and potential for further applications. This structured approach allows not only for the presentation of results but also for providing a comprehensive overview of the advantages and limitations of the proposed platform in the context of existing approaches.

### 3.1. Verification of the Measuring Device

Before the main experimental phase, verification of the proposed platform was carried out to confirm reliable long-term operation and the absence of systematic errors across the entire measurement chain (sensor → processing → storage → visualization). Figure 8 shows the final prototype with four parallel measuring branches immediately after commissioning. After successful filling and venting of the U-tubes, the flawless operation of the differential pressure sensors, reliable switching of the 3/2 valves, and functional data acquisition to the cloud storage were confirmed. These steps demonstrate the complete implementation of the solution and its readiness for metrological verification and comparative measurements.

A key feature of the design is the achieved robustness while maintaining structural simplicity, which was evident already during prototype assembly. Preparing the design in CAD software with clearly defined positions of all components enabled intuitive and precise assembly without the need for post-adjustments. Since the project is conceived as an open-source solution, only standardized and easily accessible components were used (aluminum profile frame, transparent panels, quick couplings, and hoses of common dimensions). These attributes support direct replication of the measurement device in other laboratories without the need for specialized manufacturing facilities. Measurement integrity and serviceability are reinforced by the physical separation of gas/water lines from electrical wiring, which is routed through cable ducts and feedthroughs, minimizing the risk of unwanted interference with the measuring geometry.

The distribution cabinet (Figure 9) integrates a common 12 V/5 V power supply, relay modules for valves, peristaltic pumps, and stirrers, dedicated PWM MOSFET drivers (Eclipsera s.r.o., Havlíčkův Brod, Czech Republic) for heating, and the main control unit. The layout ensures strict separation of power and signal lines; wiring is routed through cable ducts, and terminal blocks are used exclusively for power distribution, while I/O connections are logically grouped and labeled according to the documentation. To minimize interference and signal path length, the I^2^C multiplexer is located in the lower compartment directly next to the pressure sensors, so only a single short I^2^C line leads to the cabinet. Shielded cables are used for I^2^C and temperature sensors to suppress crosstalk from power circuits. This arrangement supports both mechanical and electrical stability of the measurement chain during long-term operation.

During the initial integration, brief communication dropouts on the I^2^C bus and occasional controller restarts were observed, temporally correlated with the switching of 3/2 valves and peristaltic pumps. The cause was identified as inductive spikes from solenoids and commutators. The issue was resolved by adding flyback diodes connected antiparallel directly across the terminals of each valve and DC motor, placed as close as possible to the source of interference. After the modification, transient spikes on the 12 V branch no longer propagated into the logic section, and long-term recordings contained no artifacts related to actuator switching; communication with individual peripherals proceeded without errors, confirming the electrical stability of the solution.

To verify the long-term reliability of the measurement chain, a 24 h stability test was performed. One measuring branch of the U-tube was pressurized to a fixed value and then hermetically sealed (no gas flow). The equivalent volume of the water column was continuously computed from the measured differential pressure. The measurement record (see Figure 10) can be divided into three phases: initial stabilization 0–5 h, a transient phase 6–10 h, and a long steady segment 11–24 h. The transient behavior in the 6–10 h interval is interpreted as spontaneous thermodynamic relaxation of the system and mechanical settling of pressure ratios in elastic components after closing the measuring branch. The overall range over 24 h was 28.70–29.66 mL, with this width determined primarily by the transient changes.

Statistical analysis confirmed excellent stability after stabilization. In the initial stable phase 0–5 h, the mean volume was *μ* = 29.045 mL with a standard deviation σ = 0.064 mL. The linear trend, computed by the ordinary least-squares method (*v(t) = a + bt*), corresponded to a drift of −0.027 mL·h^−1^, i.e., a total decrease of ≈ −0.14 mL over 5 h. Even higher stability was demonstrated in the longer steady phase 11–24 h, where the mean volume was *μ* = 29.004 mL and the standard deviation *σ* = 0.062 mL. The drift in this long period decreased significantly to −0.003 mL·h^−1^, i.e., only ≈−0.039 mL over 13 h.

The observed consistent short-term scatter (~0.06 mL) and the very low long-term drift (<0.15%/24 h at a nominal volume of ~29 mL) clearly confirm that, once conditions have stabilized, the entire measurement chain is stable, reproducible, and suitable for long experimental runs without the need for ongoing zero-point correction or local filtering. For illustration of accuracy, the 95% confidence interval of the mean in the steady phase 11–24 h was ±0.003 mL, and the empirical quantile range 2.5–97.5% was 28.898–29.193 mL. For visualization purposes, the transient behavior within 6–10 h may be suppressed by a mild approximating curve; however, all primary metric calculations (mean, variability, drift) were determined from raw samples.

To ensure traceability of the pressure–height–volume conversion, the geometry of the measuring U-tube (8 mm inner diameter) was verified at several points using a digital caliper, with deviations below 0.1 mm. The sensor response was validated by stepwise injection of known reference volumes. A laboratory syringe with 0.1 mL graduation was used to dose water in 1–10 mL increments. Although syringe dosing introduces minor manual uncertainty, it provides a sufficiently accurate reference for verifying the response of the system. For each injected step, the corresponding differential-pressure change was recorded and converted back to volume. No measurable hysteresis was observed between ascending and descending sequences, and deviation from ideal linearity remained within ~2% across the operating range.

Volumes in this work are expressed in mL under the laboratory ambient conditions used throughout the experiments. The term “STP” in the figures refers to these internal reference conditions (≈24 °C, 1 atm), which were kept stable over the entire measuring period. Because a differential pressure sensor is used, barometric pressure does not influence the measurement. No correction for water-vapor partial pressure was applied, as no condensation was observed in the measuring hoses.

Inter-branch equivalence was evaluated using a repeated 5 mL step test carried out on all four parallel measurement branches. The coefficient of variation (*CV*) of reconstructed step volumes among branches was 9.0%, which lies at the lower end of typical intra-laboratory variability reported for batch anaerobic digestion assays (commonly 10–20%). This indicates that the observed differences are dominated by the natural heterogeneity of the anaerobic sludge rather than by inconsistencies in the measuring branches [8].

The long-term stability of the temperature chamber, which is crucial for maintaining reproducible experimental conditions, was verified through a 90 h test at a setpoint of 37 °C under PID control. The temperature was continuously sampled and evaluated from raw data without any local filtering. The recorded temperature profile (Figure 11) exhibits a typical steady-state regime with short, symmetric corrections around the setpoint corresponding to heater switching within the controller’s dead zone. The total peak-to-peak variation over 90 h was only 0.12 °C (min = 37.00 °C, max = 37.12 °C), with 95% of all samples lying within a very narrow interval of 37.070–37.130 °C.

Statistical analysis confirmed the high reproducibility of the regulation. The mean temperature over the entire period was *μ* = 37.099 °C with a standard deviation *σ* = 0.019 °C. The linear trend, estimated by the ordinary least squares method (*T(t) = a + bt*), yielded a drift coefficient (*b* = −1.0 × 10^−4^ °C·h^−1^), corresponding to a negligible total drift of approximately −0.009 °C over the full 90 h. Although the root-mean-square error (RMSE) from the setpoint was 0.103 °C, the slight mean shift of approximately +0.10 °C reflects the natural equilibrium temperature of the closed chamber under continuous heating, not a sensor bias, since the individual offsets of both NTC sensors were already corrected in software prior to the experiment. The 95% confidence interval of the mean is exceptionally narrow 37.0980–37.0990 °C, confirming a very precise determination of the average temperature.

The observed short-term variability of ≈ ± 0.02 °C (*σ*) and the virtually zero long-term drift clearly confirm that, once stabilized, the temperature control is steady and suitable for long experimental runs without any need for additional software correction. The control loop employs a discrete PID algorithm with heuristically tuned parameters (*Kp* = 2.0, *Ki* = 0.4 s^−1^, *Kd* = 1.2 s) and implemented anti-windup logic. Such tuning effectively prevents oscillations and ensures short, symmetric corrections around the target temperature. From the perspective of maintaining constant conditions for biogas-producing reactors, the temperature loop therefore does not represent a limiting factor.

For clarity, it should be noted that the temperature shown in Figure 11 represents the output of the corrected NTC sensors described in Section 2.3 (KY-013 modules, individually offset-compensated after dry-chamber calibration). The observed *σ* = 0.019 °C therefore reflects the thermal stability of the chamber under PID control rather than the intrinsic accuracy of the sensors. A fixed software correction (+0.11 °C and +0.18 °C for the two modules) was applied before the experiment, and no additional offset compensation or filtering was used during the 90 h stability test.

Control of the entire device and remote experiment monitoring are implemented through a cloud interface (Arduino Cloud), which provides full connectivity and control. The interface is accessible directly via a standard web browser as well as through a dedicated mobile application. It is designed as a responsive user interface (UI), ensuring equal and convenient usability of control elements and graphs regardless of platform (desktop or smartphone).

Although the presented device is intended primarily as a research prototype, the chosen cloud platform implements standard security mechanisms to ensure safe data transmission. All communication between the microcontroller and the cloud is handled through a TLS-secured MQTT broker with full TLS 1.2 encryption and server-side certificate validation, preventing interception or manipulation of transmitted data. Device authentication relies on unique API keys bound to the user account, and access to dashboards requires HTTPS-based user authentication. Historical datasets stored in the cloud remain accessible only within the associated private workspace; no data are publicly visible without explicit sharing [11].

The user interface (see Figure 12) is designed pragmatically to minimize operational effort. For each reactor unit, key control actions are grouped together, specifically measurement start, experiment reset, and switching to manual mode. The main process parameters, such as the temperature setpoint of the incubation chamber, are always displayed alongside the currently measured value, ensuring continuous operator awareness.

Biogas visualization is dual-channel: for each reactor, cumulative biogas production and production aggregated over the last 24 h are displayed simultaneously. The graphs are interactive and support dynamic time-window selection (e.g., LIVE/1 h/1 d/7 d), zooming, and panning through historical data. Historical records are continuously stored in the cloud and can be easily exported in CSV format for subsequent statistical analysis. It is important to note that all displayed numerical and graphical values are derived directly from the raw data stored in the cloud; the visualization applies no local filtering or modification of measurement data, ensuring full transparency and data integrity.

### 3.2. Dataset from Automated Measurement

In accordance with the experimental protocol (2.7), four bench reactors were started in parallel: three substrate replicates (glucose, 1 g in 160 mL of anaerobic sludge) and one blank without substrate. The platform continuously evaluated the equivalent volume from the hydrostatic pressure in the U-tube and sent to the cloud the cumulative volume (mL, STP), the 24 h increment (mL/24 h), and the chamber temperature (°C). The platform samples every second; no local filtering or smoothing was applied. For graphical presentation in this section, we use a deterministically downsampled 1 min version of the same records. The downsampling is performed by deterministic decimation: within each 60 s window, the last valid raw sample is selected without any interpolation, averaging, or smoothing. This approach reduces the size of the dataset and improves the readability of long time-series plots while preserving all geometric features of the production curve, including breakpoints, slope changes, and transient segments. No quantitative analysis (e.g., drift estimation, statistical testing, or production normalization) is based on the downsampled data; all numerical evaluations rely strictly on the full 1 Hz dataset, which remains available in the cloud for verification and traceability.

An overview of the entire 0–90 h cycle (Figure 13) shows that all three substrate replicates (R_1_–R_3_) reached a stable plateau after glucose depletion, with no systematic drift observed. The control branch (blank) represents only a low system background. This long view is key for comparing final yields and assessing long-term stability after the process has stabilized.

The complementary 0–24 h detail (Figure 14) highlights the initial dynamics: first a short lag phase (acclimatization of the microbial consortium and dissolution of initial CO_2_ in the aqueous phase), followed by a sharp increase in production as acidogenic and methanogenic fluxes become established.

With glucose as an easily degradable substrate, the transition to the growth phase is sharply defined. Thanks to minute-level granularity without ex-post interpolation, the system allows precise determination of both the onset time and the slope of kinetic curves. The offsets between replicates R_1_–R_3_ are small yet stable and reproducible, confirming the high resolution of the platform in capturing subtle kinetic differences. These data serve as the basis for metrological evaluation and cross-comparison with manual reference methods (Section 3.4).

To quantify biological and measurement variability across parallel reactors, cumulative biogas production was evaluated for all three substrate reactors (R_1_–R_3_) at 24 h intervals (Table 2). The first 24 h exhibited the largest dispersion, reflecting natural heterogeneity of the inoculum and early-stage metabolic dynamics (*mean* = 155.08 mL, *SD* = 13.67 mL, *CV* = 8.81%). In later intervals (24–48 h and beyond), all reactors converged to a plateau, and daily increments approached zero, leading to vanishing variance and undefined confidence intervals.

The final cumulative production after 90 h reached 158.49, 169.12 and 141.19 mL for reactors R_1_–R_3_, respectively. After subtracting the blank correction (0.96 mL), the resulting mean was 155.31 mL. The spread between the three substrate reactors reflects genuine biological variability—primarily differences in the micro-scale composition of the inoculum and the distribution of degradable solids—rather than limitations of the measuring system. Since each reactor was filled with sludge manually, slight variations in the proportion of active biomass and particulate content are unavoidable and naturally translate into differences in the initial metabolic activity and total gas yield [8].

A small residual volume (0.96 mL over the 90 h experiment) was observed in the blank branch, despite the absence of added substrate. This behavior is fully consistent with the expected physicochemical dynamics of anaerobic sludge and water-filled measuring systems. In particular, the blank reactor still contains inoculum, which typically releases a minor amount of dissolved CO_2_ during the first hours after warming to mesophilic temperature. According to Henry’s law, the solubility of CO_2_ in water decreases substantially—by roughly 30–40%—when heating from typical room temperature (~20 °C) to 37 °C. For a liquid volume of 160 mL and an initial dissolved CO_2_ concentration on the order of 1·10^−3^ mol·L^−1^ (≈40–50 mg·L^−1^), this corresponds to the release of approximately 1 mL CO_2_ at STP, once the system reaches thermal equilibrium. Such an order-of-magnitude estimate aligns very closely with the measured 0.96 mL and strongly supports the interpretation that the blank signal originates solely from physical degassing rather than microbial activity [6,8]. Moreover, the absence of any continued increase after the first 24 h indicates that the background effect is transient, self-limiting, and negligible relative to the biogas production observed in the substrate-loaded reactors. This confirms that the measurement system itself does not contribute a systematic gas signal, and that the blank behavior reflects only the expected thermal release of dissolved gases entrained in the inoculum.

### 3.3. Dataset from Manual Methods

As part of the reference evaluation, three substrate reactors and one blank test were monitored in parallel using the discrete pressure method. The cumulative gas volume was determined at predefined time points and subsequently corrected for the blank production. The resulting curves are mutually consistent, transition to a plateau after substrate depletion, and show low variability among replicates, confirming the reproducibility of the procedure and the reliability of final yield determination (Figure 15).

In discrete data collection, however, the initial hours are typically characterized by a stepwise profile: longer intervals between readings aggregate the early phenomena (lag phase and onset of the growth phase) into a few points. From such a record, it is therefore not possible to accurately derive the instantaneous gas production rate or the exact onset time. Nevertheless, the reference dataset provides a robust basis for interpreting the process at the level of final cumulative volumes and clearly documents system stabilization after substrate depletion.

After 90 h, the reactors with substrate reached cumulative volumes of *V_R_*_1_ = 156.0 mL, *V_R_*_2_ = 142.5 mL, and *V_R_*_3_ = 151.0 mL, while the control branch (blank) measured 3.0 mL. After subtracting the blank, the mean of the replicates was 146.8 mL.

### 3.4. Comparison of Automated and Manual Approaches

The comparison is based on parallel experiments with glucose conducted according to the same protocol (Figure 16). The automated platform recorded data continuously with one-second sampling and without local filtering; for graphical presentation, the time series were deterministically downsampled to one-minute intervals without altering the metrics. The manual pressure reference was performed discretely at predetermined times of 19 h, 24 h, 43 h, 66 h, and 90 h. The difference in temporal granularity is crucial for interpretation: continuous recording allows integration of the initial dynamics, including the ramp-up from the lag phase, and enables precise identification of the transition to the plateau, whereas discrete readings aggregate phenomena between points and provide no information on instantaneous rates during the early hours. It should be noted that the labels R_1_–R_3_ serve only as local identifiers within each method; R_1_, R_2_, and R_3_ in the automated measurement do not correspond to the same physical reactors as R_1_, R_2_, and R_3_ in the manual measurement.

The shape of the curves is qualitatively identical in both approaches: a low background in the control branch, a steep onset of production after microbial acclimatization, and subsequent stabilization after substrate depletion. Continuous data additionally reveal subtle kinetic nuances between replicates without the need for ex-post interpolation, which is advantageous for modeling and retrospective evaluation. Discrete measurement provides robust final volumes; however, the stepwise profile within the 19–24 h interval naturally merges key transient phenomena into a few points, limiting the derivation of rate metrics.

After 90 h (Table 3), the reactors with substrate under automated measurement reached cumulative volumes of *V_R_*_1_ = 158.49 mL, *V_R_*_2_ = 169.12 mL, and *V_R_*_3_ = 141.19 mL; the control branch measured 0.96 mL. After subtracting the control, the mean of the replicates is 155.31 mL. Manual measurement after 90 h, corrected for the control at the same time, yielded 153.0 mL for *V_R_*_1_, 148.0 mL for *V_R_*_2_, and 139.5 mL for *V_R_*_3_; the mean of the replicates is 146.83 mL. The difference between the means is 8.47 mL. This difference is expected when comparing continuous integration with sampling at sparse time points: short but volumetrically significant ramp-up segments are fully captured by continuous integration, whereas in manual measurements they appear only at the nearest reading point and may lead to a lower final total.

Using the three reactor pairs as independent replicates (*N* = 3), the mean relative bias between the automated and manual methods was 5.78%. Due to the limited number of replicates, the corresponding 95% confidence interval was wide (−11.6% to 23.0%), and the bias was not statistically different from zero (paired *t*-test, *p* = 0.29). This indicates that the observed difference remains small compared to the biological variability typically encountered in BMP assays and does not alter the practical agreement between both methods.

The difference is also influenced by the method of handling. Continuous measurement is performed in a closed system without reactor intervention, minimizing undesired effects on pressure and thermal equilibrium. In contrast, manual measurement requires operator intervention, manipulation with sampling tools, and temporary removal from the incubation environment, which introduces short-term temperature and pressure deviations, possible micro-leakages during handling, and the need for optical reading from the scale. Combined with the fact that the entire interval between readings is integrated without information about its internal progression, there is a natural tendency toward slightly lower final volumes in manual records.

The background of the control branch remained low and stable. In the manual measurements, it reached 3 mL after 90 h and was consistently used for replicate correction. In the continuous data, the control remained near zero throughout the experiment and did not represent a practical limit for interpretation. Blank correction does not alter the qualitative conclusion: the results of both approaches are consistent in terms of final yields, while the continuous measurement additionally provides a time-dense kinetic profile necessary for accurate derivation of temporal metrics and metrological evaluation.

### 3.5. Metrological Aspects of Automated Measurement

Measurement of biogas production is based on the hydrostatic principle, where the generated gas displaces water in the measuring branch of a U-tube, and the change in differential pressure Δ*p* (Pa) is directly converted into the level displacement Δ*h* (mm) and subsequently into volume *V* (mL). The theoretical conversion follows the relationships:(4)$$\Delta p\;=\rho_{w}\cdot g\cdot \Delta h\;\;\;\Rightarrow \;\;V\;=A\cdot \Delta h\;=\frac{A}{\rho_{w}\;\cdot \;g}\;\Delta p\;\;\Rightarrow \;\;V=\;\frac{\Delta p\;\cdot \;\pi \;\cdot \;d^{2}}{4\;\cdot \;\rho_{w}\;\cdot \;\;g}$$
where $\rho_{w}$ is the water density ($kg\cdot m^{-3}$), g is the gravitational acceleration ($m\cdot s^{-2}$), and A  is the effective cross-sectional area of the measuring branch ($mm^{2}$). Under laboratory conditions (T ≈ 22 °C), a water density of $\rho_{w}\approx 997.585\;kg\cdot m^{-3}$ and gravitational acceleration $g=9.80665\;m\cdot s^{-2}$ are considered. During the measurement, PID temperature control is active only in the upper chamber containing the reactors, while the U-tube and pressure sensor are located in a climate-controlled laboratory with a typical temperature fluctuation of ±1 °C. For an inner branch diameter D=8 mm, the effective cross-sectional area is $A\approx 5.026\times 10^{-5}m^{2}$.

Based on these parameters, the pressure-to-volume sensitivity coefficient ($c_{V}$) is used for consistent conversion of pressure uncertainties to volumetric uncertainties [26]:(5)$$c_{V}=\frac{\partial V}{\partial p}=\frac{A}{\rho_{w}\cdot g}\approx 5.137\times 10^{-9}\;m^{3}\;\cdot Pa^{-1}$$

From this, it follows that 1 mbar corresponds to a volume V≈0.514 mL (1 mbar ≈ 100 Pa). This constant is used for consistent conversion of pressure uncertainties into volumetric ones.

The main influencing parameters of the measurement and their estimated uncertainties (determined by Type A and Type B methods) [27] are summarized in the uncertainty balance Table 4. The values are established for the prototype of the automated platform and are derived from calculations, documentation, or qualified estimates.

Justification and detailed interpretation of the items:

Repeatability (item 1) was estimated from short-term variance under constant conditions. For a sensor full scale FS = 120 mbar and a specification of 0.1% FS, the absolute uncertainty is 0.12 mbar, which, after conversion using the sensitivity $c_{V}$, corresponds to 0.062 mL. This component is absolute (it derives from measured data, accounting for the sensors’ technical documentation) and decreases relative to the measured volume. For example, for a single step ΔV=5mL, this component accounts for 1.24%, whereas at a cumulative volume of 100 mL it drops to approximately 0.062%.Time granularity of the record (item 2) represents the discretization error of the integral. Acquisition at 1 Hz oversamples the dominant phenomena (time constants ≫1 s), making its influence small. The upper bound of the trapezoidal-integration error is on the order of Δt/2⋅max∣dQ/dt∣. The estimate of this component is 0.8%, with a shorter step and a guard window after switching reducing aliasing of transients.Integration of the initial dynamics (item 3) concerns the most sensitive segment of the measurement—the ramp-up from the lag phase. If the moment of ramp-up is uncertain by δt, it causes a volumetric deviation of approximately $R_{max}\cdot \delta t$. With continuous recording and a short guard window, the uncertainty δt is small. A value of 0.5% is retained as residual risk, which includes delay and smoothing of the derivative.Evaluation method (item 4) includes temperature/pressure corrections, choice of integration, and rules for outliers. Differences among reasonable choices are typically below 1%, therefore 1.0% is estimated conservatively. The key is transparent, deterministic data processing (the same rules for all runs) [27].Physical conditions in the measuring section (item 5) are determined by room-temperature fluctuations (±1 °C). A temperature change of ±1 °C alters the density of water around 22 °C by only ≈0.03% per 1 K. More significant are secondary effects such as gas solubility and media expansion. With reasonable compensation (reference to $p_{atm}$, stable water interface), this component is kept below 0.5%. Also considered are atmospheric pressure, humidity, and gravitational acceleration. These quantities are directly metrologically traceable to calibrated instruments of the respective measurands, were measured, and were included in the balance table.Other influences (item 6) include joint tightness, microbubbles, wall wetting, and biofilm, which affect the effective cross-section A. These influences are managed by protocols (leak test, cleaning, standard filling). Also included are other system-wide effects which, in further development of covariances, may be mutually independent sources of uncertainty and thus contribute to the resulting uncertainty. For this reason, it is important to perform more experimental measurements on the prototype of the automated platform for measuring biogas production and, in the future, to address covariances that can increase—but also significantly decrease—the resulting uncertainty. Everything depends on whether the uncertainty components act concordantly or discordantly on the two considered uncertainty estimates; the influence of the function tied to the output quantity is also important.

These components represent the metrological characteristics of the platform. Covariances may exist among individual uncertainty estimates, and their quantification can increase or decrease the resulting uncertainty depending on the nature of the couplings as outlined above.

To quantify the final system performance, the combined standard uncertainty $u_{c}$ (mL) was calculated by the root-sum-of-squares [26] method (assuming independence of components) for a typical cumulative volume $V_{cum}=100\;mL.$(6)$$u_{c}=\sqrt{u_{A}^{2}+u_{B1}^{2}+u_{B2}^{2}+u_{B3}^{2}+u_{B4}^{2}+u_{B5}^{2}}$$(7)$$u_{c}=\sqrt{2.393844}\approx 1.547mL$$

Including all six components yields $u_{c}\approx 1.55\;mL.$ Based on this result and using an expansion coefficient k=2 (≈95% interval), [26] the expanded uncertainty is U≈3.09 mL, i.e., ≈3.09% of 100 mL.(8)$$U=k{\;\cdot u}_{c}$$(9)U=2 ⋅1.547 mL≈3.09mL

These quantitative results and the uncertainty balance confirm the robustness and suitability of the system for long-term, precise measurement of biogas production. With an expanded uncertainty U=3.09 mL for a typical cumulative volume of 100 mL, the system achieves metrological performance comparable to commercial reference devices, providing a low-cost and replicable platform for the global scientific community. An expansion coefficient k=2 was used, which, for a normal distribution, corresponds to a confidence probability of approximately 95%.

## 4. Discussion

The proposed low-cost platform demonstrates the metrological stability necessary for advanced analysis of anaerobic digestion (AD) kinetics. Our results (Section 3.1) confirm strict temperature control (σ ≈ ±0.02 °C) and negligible long-term drift of the zero baseline, thereby eliminating environmental conditions as a source of uncertainty. This stability meets, and in some aspects exceeds, the minimum requirements for laboratory biomethane potential (BMP) tests [28,29]. A key contribution is the verification of the hydrostatic principle, which places the platform within the new wave of Low-Cost IoT solutions using ESP32 microcontrollers for efficient bioreactor monitoring [7,30].

The stability of the prototype was assessed using three complementary datasets that capture thermal, mechanical, and operational behavior. Long-term thermal stability was characterized through the 90 h chamber test, where variance and drift analysis demonstrated that temperature fluctuations and baseline shifts remained far below the threshold that would influence gas–liquid equilibrium. Mechanical and sensing stability were evaluated through repeated step-injection experiments, which confirmed a consistent pressure-to-volume conversion, negligible hysteresis during ascent/descent cycles, and reproducibility across the four measurement branches. Finally, operational stability was assessed using biological replicates (R_1_–R_3_), for which mean values, standard deviations, and confidence intervals were computed. These results show that the residual spread in cumulative volume is governed by biological variability rather than instrumental noise. Together, these analyses demonstrate that the device maintains stable metrological performance under thermal, mechanical, and real-operation conditions, validating its suitability for long-term anaerobic digestion monitoring.

When benchmarked against existing biogas-monitoring platforms, the proposed system offers a performance profile that is comparable to commercial hydrostatic or manometric devices and, in several parameters, exceeds the capabilities of other low-cost solutions. The short-term precision observed in the sealed-branch stability test (*σ* ≈ 0.06 mL) matches the repeatability typically reported for laboratory-grade differential-pressure instruments. The long-term drift (<0.15% per 24 h) is also within the range of commercial automated volumetric systems and substantially better than what has been documented for low-cost manometric or headspace-accumulation devices, which are more susceptible to temperature fluctuations and membrane leakage. The 1 Hz sampling rate provides two to three orders of magnitude higher temporal resolution than common BMP analyzers (5–15 min sampling), making it possible to capture transient kinetic features that are normally lost in coarse-resolution systems. Together, these characteristics position the platform between high-end commercial analyzers and low-cost academic prototypes: it achieves metrological performance close to the former while maintaining the affordability and openness of the latter.

The 5.78% difference in average final yield (Table 3) between the continuous and manual approaches is a critical point of discussion, and we consider this figure a direct quantification of the methodological limitations of discrete methods, not an error of our continuous system. The manual method (MM) has two key weaknesses. First, sparse sampling cannot accurately integrate the biogas volume, especially during short segments of steep production ramp-up. Second, repeated handling of the reactors during overpressure release and reading introduces pressure and thermal shocks that lead to gas losses. This handling error is also confirmed in the literature. Pérez-Vidal et al. (2025) even quantified an average loss of 50.7 ± 12.9 mbar per measurement with the MM approach, while this loss was eliminated in their continuous system [7]. Moreover, Holliger et al. (2016) have long emphasized that non-standardized protocols and operator interventions are the main cause of poor inter-laboratory reproducibility of BMP tests [28,29]. Our continuous solution addresses both methodological weaknesses by eliminating the human factor and providing high-granularity data.

The high temporal resolution (second-level granularity) of our platform is essential for advancing kinetic modeling of AD. Models such as the modified Gompertz model are critically dependent on the precise identification of two parameters: λ (length of the lag phase) and *R_max_* (maximum production rate) [31]. Discrete data that compress this dynamic into a few points inevitably obscure the exact timing of the ramp-up, leading to high parametric uncertainty and unstable estimates of kinetic metrics. The ability of our system to reliably quantify and stabilize the lag phase and the slope of the ramp-up curve is therefore crucial. These high-quality data are ideal for robust fitting of kinetic models and can serve as input for advanced hybrid models and machine learning (ML) approaches aimed at process optimization and minimizing operational uncertainties in real time [32,33,34].

The cloud-based architecture of the platform also opens a realistic path toward intelligent anomaly detection in future iterations. Because the system produces dense and highly structured time-series data (1 Hz pressure, volume increments, vent cycles, refill events and temperature), it naturally supports lightweight deep learning models for detecting deviations from expected behavior. Such models—typically autoencoder-based reconstruction or recurrent (LSTM-type) predictors—can run on a remote server and be triggered directly from the Arduino IoT Cloud via webhook integration. This approach would enable automatic identification of events such as leak-like pressure decay, missing refill signatures, abnormal venting dynamics or unexpected shifts in gas-production trends. Importantly, the implementation does not require any hardware modification; the ML layer operates entirely on the cloud side, using the existing telemetry stream. Integrating this alerting mechanism in future versions would substantially enhance robustness in long-term unattended experiments and aligns with the broader trend of augmenting low-cost IoT monitoring platforms with intelligent diagnostic capabilities [35,36].

In addition to deep learning-based anomaly detection, future development of the platform can leverage reinforcement learning (RL) to optimize the control strategy of the system itself. The proposed architecture already provides all necessary inputs for RL—continuous time-series of temperature, hydrostatic pressure, gas-production rate, and actuator states (heating PWM, mixing, venting cycles). In an RL formulation, these variables constitute the system “state”, while actions correspond to controllable process parameters such as heater duty cycle, mixing intensity, venting timing, or adaptive sampling frequency. The agent receives a reward signal proportional to process stability, energy efficiency, or production rate, enabling autonomous learning of an optimal control policy. Recent works in building automation have demonstrated that RL can outperform classical PID or rule-based schemes in dynamic, non-linear environments with delayed responses—conditions that closely resemble anaerobic digestion reactors. Because the present system already streams all sensor data to a cloud backend, the RL agent can be trained and deployed entirely off-device, sending high-level control commands back to the microcontroller via secure cloud APIs [35,36]. This architecture eliminates computational constraints of embedded hardware and opens the possibility for self-optimizing, adaptive control of digestion processes, where the platform autonomously tunes heating, venting, and mixing strategies to maximize stability and gas-production efficiency over long-term experiments.

The proposed solution is built on an open and low-cost approach, which is consistent with the current trend in environmental research [7]. With a total investment of ≈€1960 for a fully functional system, our platform represents a radically cheaper alternative to high-end commercial systems such as the BPC AMPTS III [37]. Although the AMPTS III is the industrial standard, its acquisition cost (on the order of >€20,000 for a similar configuration) is unattainable for many laboratories. Our affordability—representing less than 10% of the benchmark price—is key to enabling advanced monitoring and supports global replication with open documentation. In addition, our system offers higher data density (1 Hz) compared to the AMPTS III, which typically measures at ∼5–15 min intervals [37].

To contextualize the metrological performance of the proposed hydrostatic platform within the broader landscape of BMP measurement technologies, a structured comparison was carried out across all relevant instrument classes: flow cell volumetric systems (AMPTS III) [37], manometric pressure-based devices (e.g., VELP Maxi [38] and the continuous IoT manometer by Pérez-Vidal et al., 2025 [7]), and semi-automated volumetric BMP sets (AnaeroTech BMP/Nautilus) [39]. These approaches represent the state-of-the-art in low-cost and mid-range laboratory testing and capture the full methodological spectrum from optical pulse quantification to headspace-pressure conversion and classical water-displacement geometries. As noted by Holliger et al. (2016) [28] and Hafner et al. (2019) [8], the majority of deviations in laboratory BMP measurements arise from methodological artifacts—particularly temperature-driven pressure drift, CO_2_ solubility effects in closed reactors, and operator-induced variability during manual handling. These limitations are clearly reflected in recent continuous manometric systems, such as that of Pérez-Vidal et al. (2025) [7], which exhibits a sensor uncertainty of 7.6 mbar and systematic biogas losses of 50.7±12.9 mbar per manual measurement. Table 5 summarizes the essential metrological and operational characteristics of these systems, allowing direct parameter-level benchmarking against the proposed hydrostatic method.

In the future, we plan to transform the first prototype into a fully featured, intelligent tool. The key development will include implementation of online gas composition analysis by integrating low-cost NDIR sensors for CH_4_/CO_2_ measurement. This will enable a complete material balance and automatic tracking of specific biomethane production [40,41]. Beyond the hardware expansion, we will focus on software intelligence: introducing automated fitting of kinetic models directly on a cloud server. The system will thus provide researchers with kinetic parameters (*λ*, *R_max_*) in real time [31]. The final step will be the implementation of internal “health checks” (e.g., zero-baseline self-test and automatic leak test) into the control software, thereby increasing reliability in long-term, unattended experiments. These steps clearly define that the platform has the potential to become a standard, low-cost, and open tool for advanced anaerobic digestion research [7].

## 5. Conclusions

This article successfully introduced and validated an open, low-cost platform for continuous monitoring of biogas production that overcomes the key methodological limitations of traditional laboratory BMP tests. The system is based on an accurate hydrostatic water-displacement principle and is controlled by an Arduino Nano microcontroller, making it a robust part of the Low-Cost IoT trend in anaerobic digestion monitoring. The statistical evaluation of the automated–manual comparison (*N* = 3, 95% *CI* = −11.6% to 23.0%, *p* = 0.29) indicates that the observed bias is statistically inconclusive and remains within the expected spread of BMP laboratory variability. The achieved long-term metrological stability, confirmed by an extremely low drift of the calibration zero and precise temperature control, demonstrates that the system’s accuracy is limited only by the actual biological variability of the process.

The key empirical contribution is the quantification of the inherent error of discrete measurement. Our continuous data showed that manual methods led to a 5.78% underestimation of the final yield. This difference is a direct consequence of elimination errors and losses caused by reactor handling and overpressure release. Our platform provides data with unparalleled temporal resolution (1 Hz), which is essential for robust identification of critical kinetic parameters such as the lag phase (*λ*) and maximum production rate *(R_max_*). This opens the possibility for reliable application of kinetic models (e.g., the modified Gompertz model) and advanced machine learning (ML) approaches for process prediction and optimization.

The current prototype reflects several design boundaries inherent to its present architecture. The system operates as a four-reactor platform, which defines its current throughput and provides the basis for future modular scaling. The thermal regulation was designed and validated for mesophilic operation ≈ 35–40 °C. While extension to thermophilic conditions is technically feasible, it would require additional validation and potential adjustments to ensure stable hydrostatic measurement under higher vapor pressure and increased condensate load. The hydrostatic principle performs reliably with standard laboratory substrates, but extreme foaming, elevated H_2_S levels, or heavy condensate formation may necessitate mechanical adaptations to maintain long-term stability. As an open-vented configuration, the platform also relies on proper laboratory ventilation and routine biogas-handling precautions to ensure safe operation.

From an economic point of view, the system represents a radical alternative to benchmark solutions. With a total investment of ≈ €1960, the platform becomes accessible to academic and research institutions worldwide, while its price represents less than 10% of the acquisition cost of high-end commercial systems (e.g., BPC AMPTS III). This affordability, together with open documentation, significantly supports global replication of the method and standardization of research. A detailed bill of materials (Appendix A) summarizes all major components together with indicative unit prices. Minor items such as fasteners, brackets and ancillary hardware are grouped as consolidated entries; their exact specifications can be derived directly from the shared CAD model and assembly documentation.

In the future, we plan to extend the platform by integrating online NDIR gas sensors for gas analysis (CH_4_/CO_2_), which will provide a complete material balance in real time. Software development will focus on automated fitting of kinetic models and implementation of intelligent self-diagnostics, making the platform a comprehensive, open, and intelligent tool for advanced anaerobic digestion research.

All datasets, firmware, wiring schematics, CAD assemblies, and analysis scripts used in this study are openly available in the public repository at Zenodo (https://doi.org/10.5281/zenodo.17674470). The repository is released under an open CC-BY license to ensure full reproducibility and facilitate reuse in future biogas-monitoring research.

## Figures and Tables

**Figure 1 sensors-25-07297-f001:**
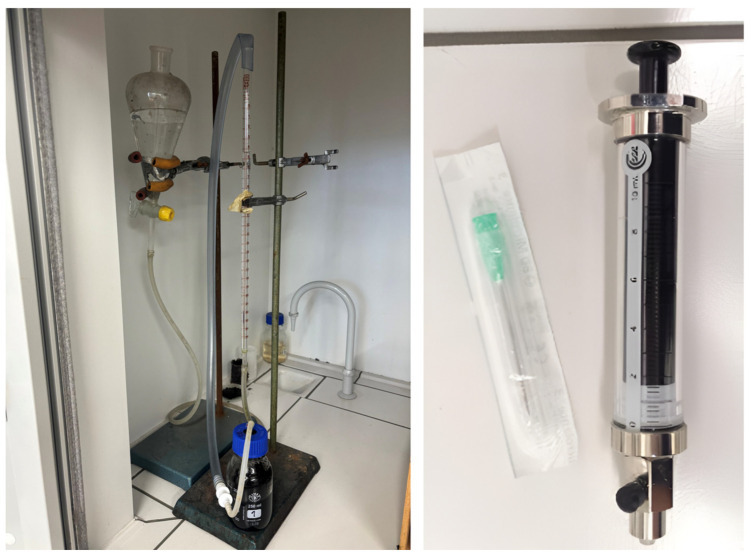
Reference manual methods used for verification. **Left**: laboratory setup for manual volumetric measurement using a water column (U-tube/measuring cylinder, water seal, and gas inlet from the reactor). **Right**: gas-tight syringe with Luer-Lock and disposable needle used for reactor sampling/connection and gas dosing during manual tests.

**Figure 2 sensors-25-07297-f002:**
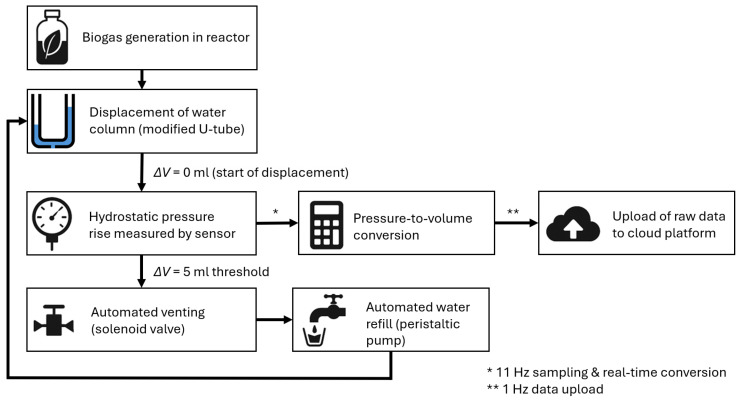
Overview of the automated hydrostatic displacement system for continuous biogas monitoring. Biogas generated in the reactor displaces the water column in a modified U-tube (Δ*V* = 0 mL at the start of each cycle). The resulting hydrostatic pressure rise is continuously measured by the sensor and converted to gas volume in real time at 11 Hz (*). Once the displacement threshold is reached (Δ*V* = 5 mL), the system triggers automated venting and subsequent water refill. Raw pressure and volume data are uploaded to the cloud platform at 1 Hz (**).

**Figure 3 sensors-25-07297-f003:**
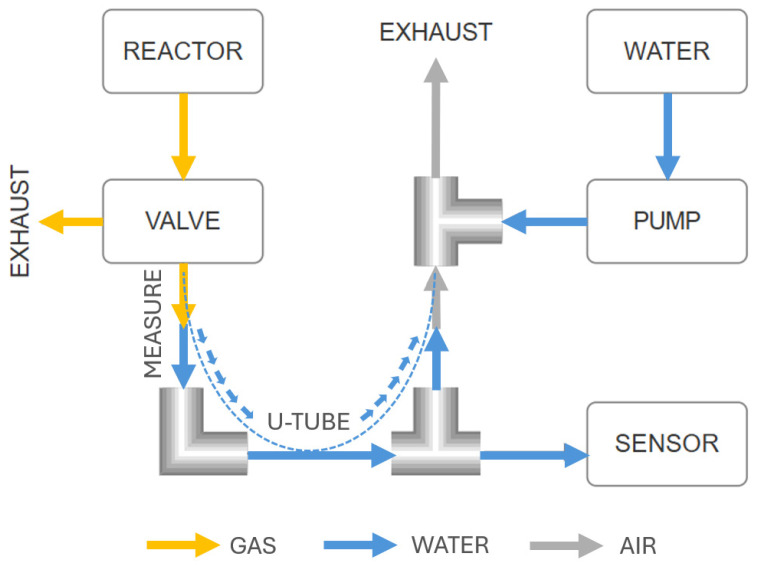
Simplified schematic of one measurement branch. Gas from the REACTOR enters a 3/2 VALVE that switches between EXHAUST and MEASURE. In measuring mode, gas flows into the U-tube; the bottom T-junction taps the pressure SENSOR (differential) to read the hydrostatic head. The top T-junction is open to EXHAUST (atmospheric reference) and laterally connected to the peristaltic PUMP for automatic water refill. Color coding: yellow = biogas, blue = water, gray = atmospheric air.

**Figure 4 sensors-25-07297-f004:**
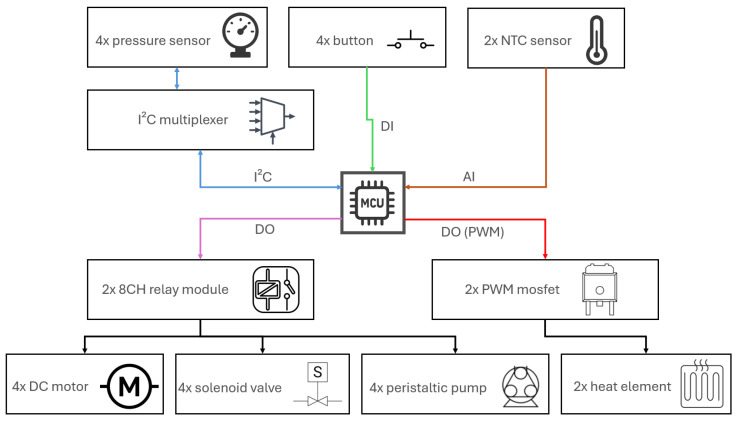
Block diagram of the control and measurement architecture. The microcontroller unit (MCU) receives pressure data from four differential pressure sensors via an I^2^C multiplexer and temperature feedback from two NTC thermistors. Digital inputs (DI) read user control buttons, while digital outputs (DO) drive two relay modules powering the solenoid valves, peristaltic pumps and DC stirrer motors. Two PWM-driven MOSFET stages supply the heat elements in the incubation chamber. All components are powered from a dedicated 360 W supply, and the MCU simultaneously manages measurement, actuation and cloud-based data transmission.

**Figure 5 sensors-25-07297-f005:**
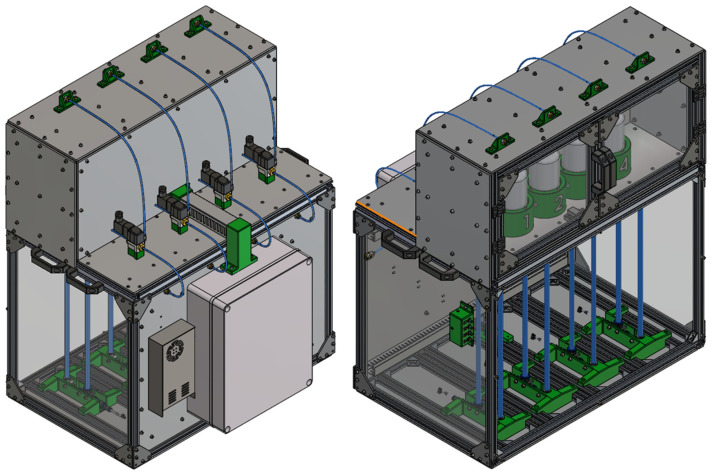
Assembly—isometric views. **Left**: rear-left view showing the 3/2 valves on the top plate, exhaust lines connected to roof quick couplings, and the placement of the electrical box/power supply; the four U-tubes are visible from below. **Right**: front-right view with transparent walls of the measuring chamber (removable front panel) and double doors of the incubation chamber; reactors are placed in numbered holders with a mirrored arrangement of the four measuring branches.

**Figure 6 sensors-25-07297-f006:**
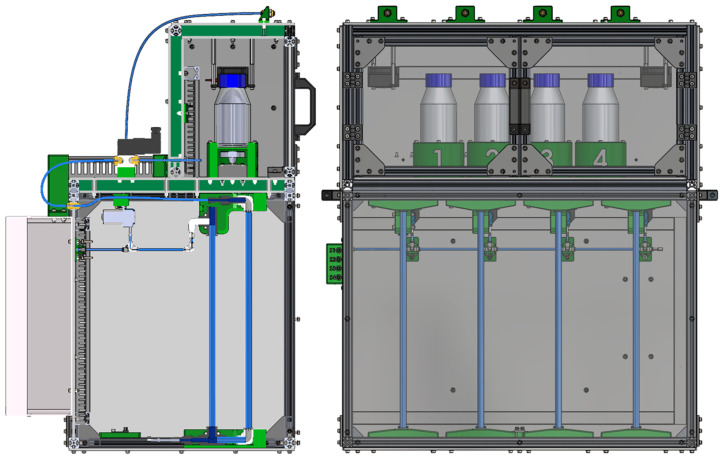
Mechanical arrangement of the biogas measurement system. On the **left** is a side section of a single measuring branch showing the reactor, 3/2-way valve, U-tube (ID 8 mm) with a reduction (ID 4 mm → ID 8 mm), connection to the differential pressure sensor, and the peristaltic pump. On the **right** is a front view of the entire device with the upper incubation chamber (four reactors) and the lower measuring chamber (four U-tubes).

**Figure 7 sensors-25-07297-f007:**
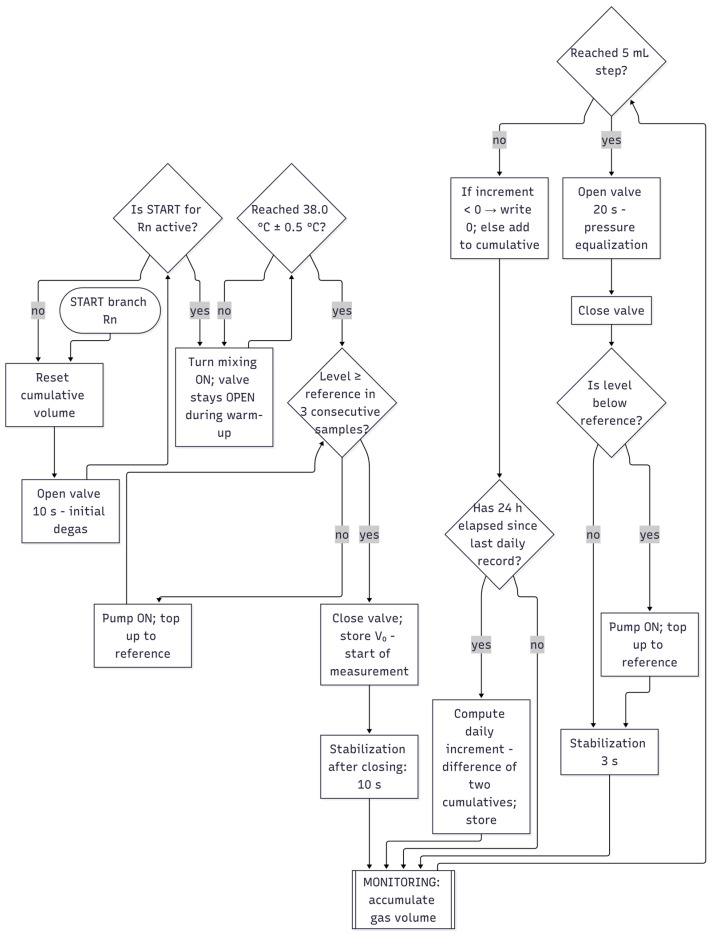
State machine of one reactor branch (R_n_). The cycle comprises four phases: accumulation, venting, top-up, and stabilization. Fixed measurement step Δ*V_step* = 5 mL keeps the sensor in its linear range and makes evaporation per step negligible. Timing: initial degas 10 s, venting 20 s, stabilization 10 s after sealing and 3 s after top-up. Reference level is confirmed by three consecutive samples (~200 ms sampling). Four identical automata (R_1_–R_4_) run asynchronously.

**Figure 8 sensors-25-07297-f008:**
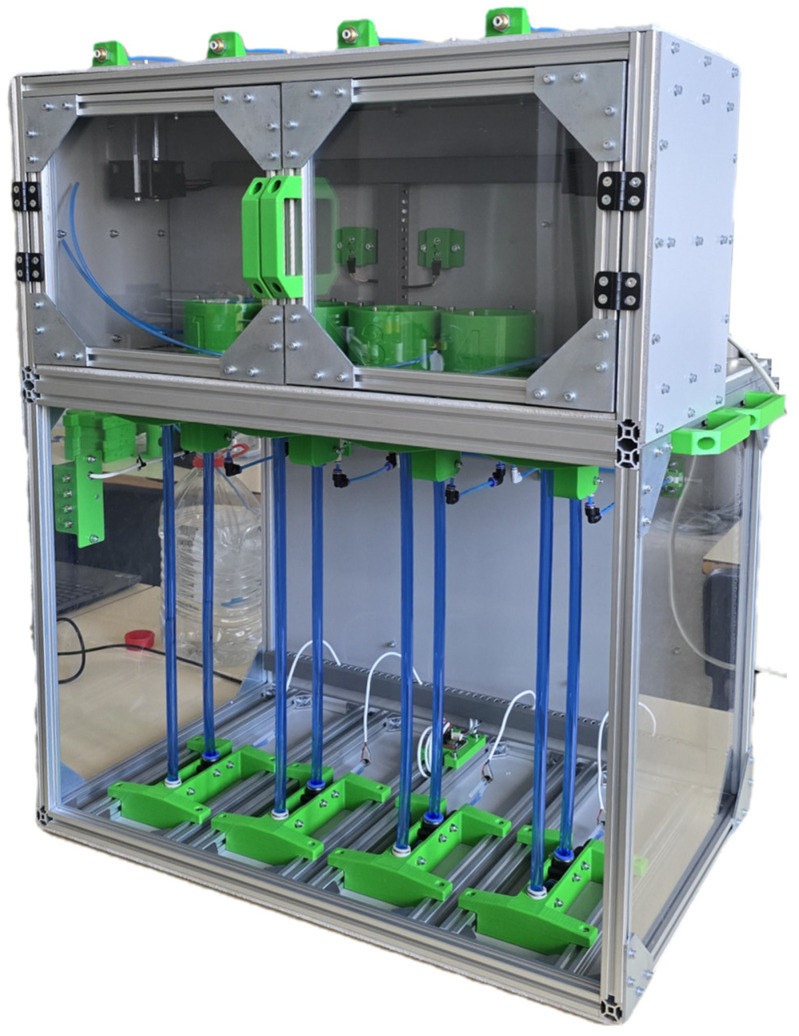
Implemented prototype of the automated platform for biogas production measurement: a dual-chamber design (upper incubation chamber with heating and mixing, lower measuring chamber with U-tubes and pressure sensors) built within a modular aluminum frame; four parallel measuring branches, transparent panels for visual inspection and maintenance.

**Figure 9 sensors-25-07297-f009:**
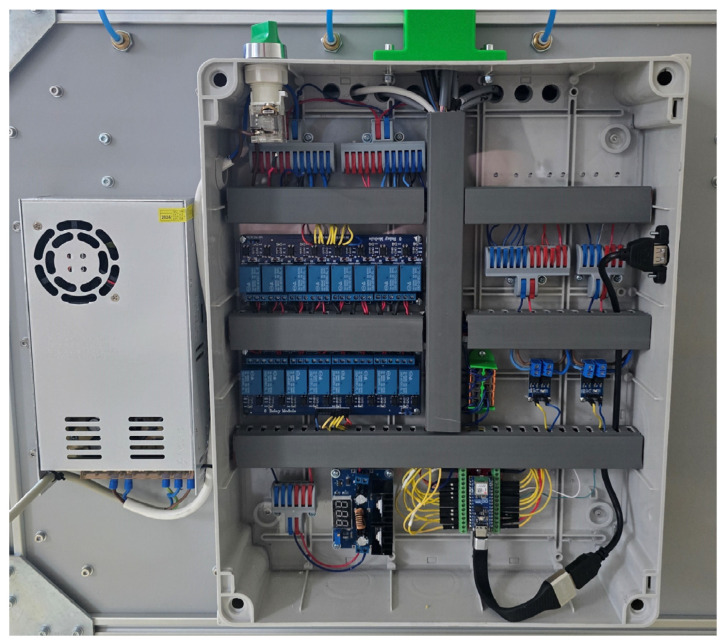
Distribution cabinet of the prototype: common 12/5 V power supply (left), relay modules for valves and peristaltic pumps, PWM MOSFET drivers for heating, and the control unit. The wiring is routed in cable ducts with careful separation of power and signal cables.

**Figure 10 sensors-25-07297-f010:**
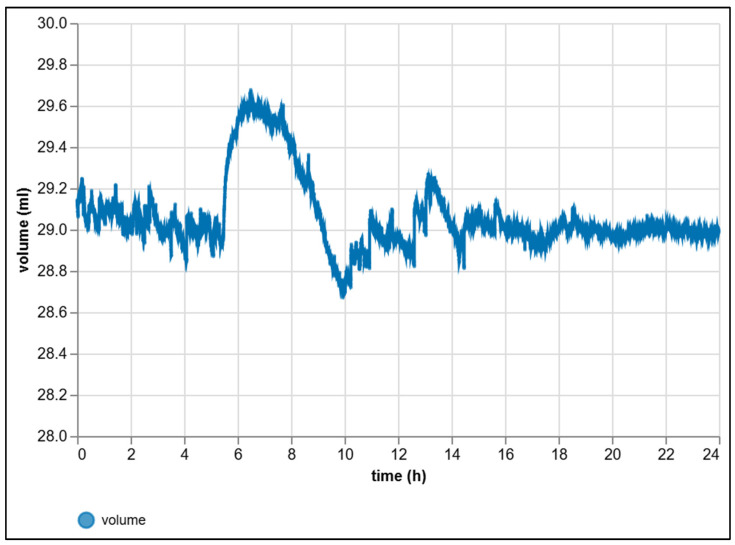
Stability test of the hydrostatic measurement with a hermetically sealed U-tube branch (no flow). Shown is the equivalent volume converted from differential pressure over 24 h: a short initial stabilization ~0–5 h, a transient phase ≈6–10 h, and a subsequent steady segment 11–24 h with a mean of ~29.00 mL, a standard deviation of ~0.06 mL, and a long-term drift of ~−0.003 mL·h^−1^.

**Figure 11 sensors-25-07297-f011:**
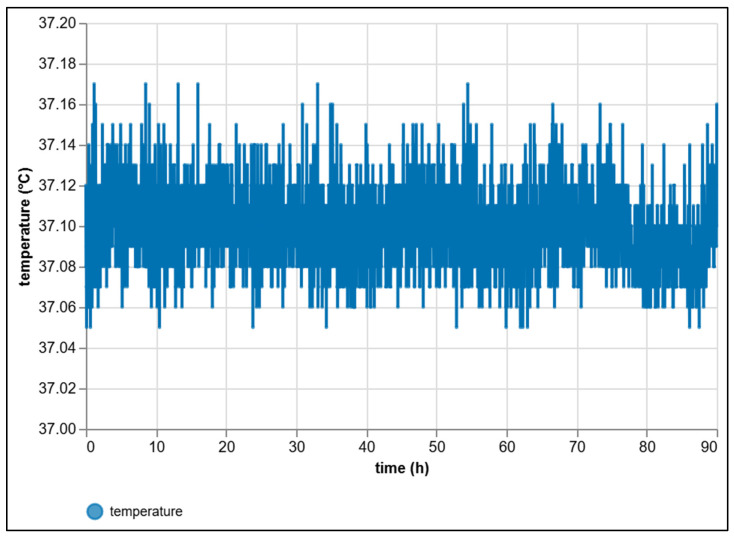
Temperature stability in the upper chamber at a 37 °C setpoint over 90 h (PID control). Mean: 37.099 °C, *σ* = 0.019 °C; 5th–95th percentile: 37.070–37.130 °C; linear drift: −1.0 × 10^−4^ °C·h^−1^ (≈−0.009 °C/90 h).

**Figure 12 sensors-25-07297-f012:**
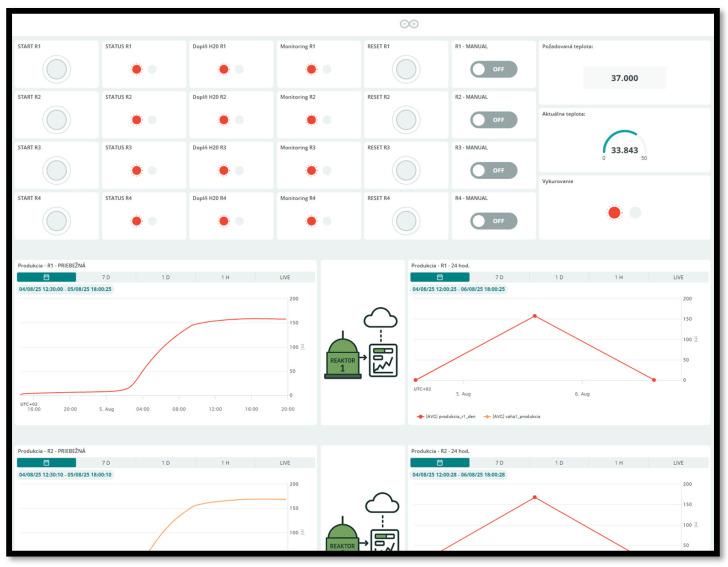
Cloud dashboard: control of individual reactors (start/reset, manual mode), setting of the desired temperature, and a pair of graphs for each reactor—continuous production and 24 h production—with support for time-interval selection, zooming, and data export.

**Figure 13 sensors-25-07297-f013:**
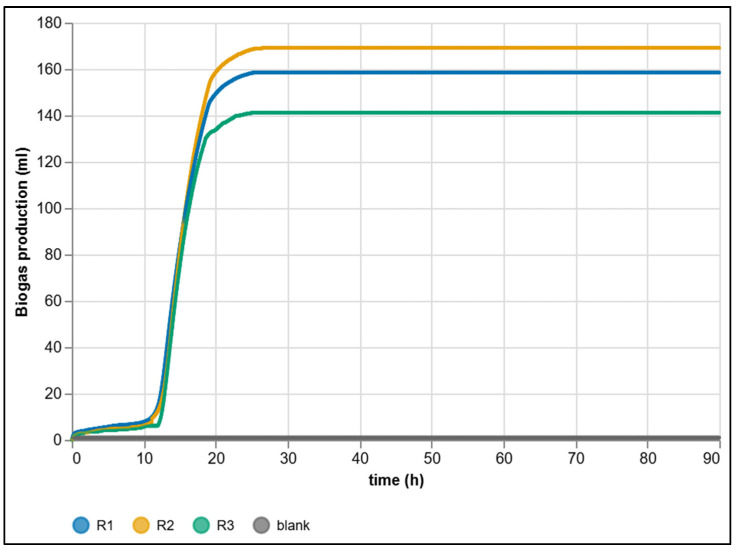
Cumulative biogas production for three parallel substrate reactors (R_1_–R_3_) and the blank at 37 °C over 0–90 h. After the ramp-up around 13–19 h, the curves reach a stable plateau (*V_R_*_1_ = 158.49 mL, *V_R_*_2_ = 169.12 mL, *V_R_*_3_ = 141.19 mL), while the blank remains ≲ 1 mL; visualization is shown at one-minute resolution.

**Figure 14 sensors-25-07297-f014:**
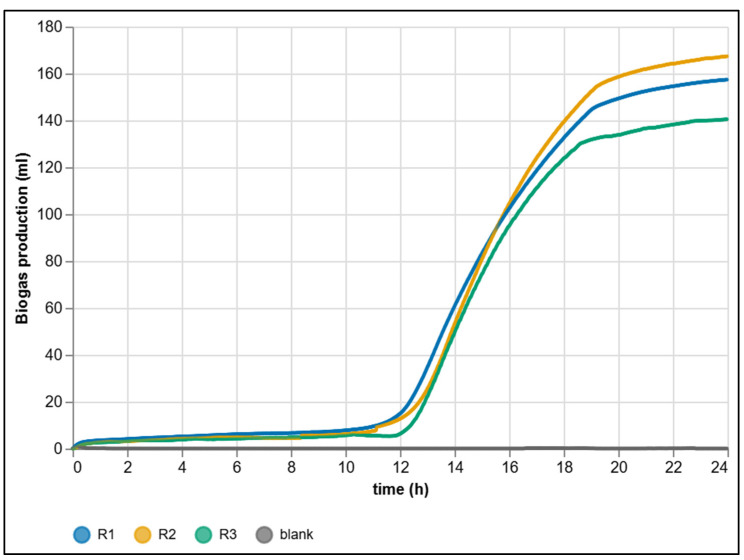
Detail of the first 24 h highlighting the initial dynamics: a short lag phase up to approximately 12–13 h, followed by a sharp increase with differing slopes among R_1_–R_3_, indicating distinct final plateaus. The control branch (blank) remains practically at zero.

**Figure 15 sensors-25-07297-f015:**
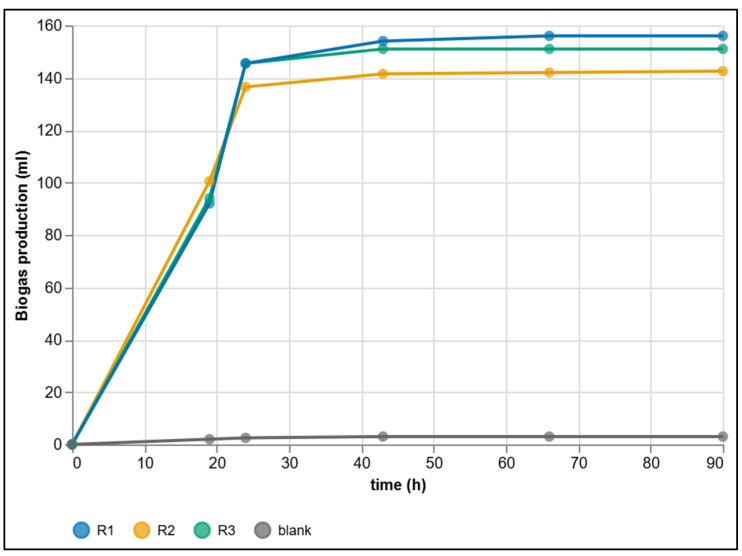
Cumulative biogas production obtained by the pressure-based (discrete) method in three parallel reactors (R_1_–R_3_) and in the blank test. The curves are composed of several readings taken at fixed times (19, 24, 43, 66, and 90) h; the final yields after 90 h reached approximately 156.0 mL (*V_R_*_1_), 142.5 mL (*V_R_*_2_), 151.0 mL (*V_R_*_3_), and 3.0 mL (blank). The coarser temporal sampling during the initial hours results in a stepwise profile and lower resolution of the initial lag phase.

**Figure 16 sensors-25-07297-f016:**
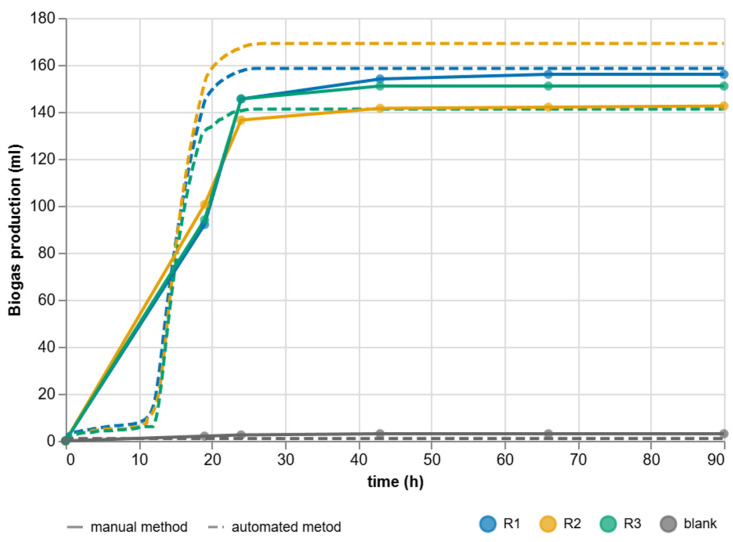
Cumulative biogas production over 90 h for three reactors (R_1_–R_3_) and the blank test. Solid lines represent manual pressure measurements; dashed lines represent automated measurements; colors correspond to individual reactors. The manual curves consist of several readings at defined time points, while the automated data capture the continuous course of the ramp-up and stabilization phases. Both approaches agree in final yields, with the main difference being the resolution of the initial dynamics. The blank test shows a low and stable background near zero.

**Table 1 sensors-25-07297-t001:** Comparison of manual and automated BMP measurement methods, summarizing key differences in time resolution, accuracy, systematic error sources, reproducibility, operational workload, and applicability in laboratory digestion experiments.

Criterion	Manual BMP Methods	Automated BMP Methods
Measurement mode	discontinuous measurements (manual readings)	continuous or semi-continuous data acquisition
Time resolution	4–24 h (depending on operator availability)	seconds to minutes (adjustable)
Data acquisition	subjective, operator-dependent	objective, automated
Measurement accuracy	±5–10%; dependent on reading frequency	stable; defined by sensor specifications and workflow
Systematic errors	negative bias, gas loss during venting, pressure drift	minimized through controlled, repeatable measurement cycles
Sensitivity to temperature fluctuations	high (gas-phase expansion/contraction)	lower (faster and more stable response)
Detection of transient events	practically impossible	captured in real time
Operator workload	high—manual manipulation and monitoring	low—automated operation
Reproducibility	low inter-lab reproducibility	high, due to defined algorithmic control
Inter-lab variability	differences may exceed 100% for identical substrates	reduced when using standardized hardware and protocol
Equipment cost	low–moderate (manual setups)	low (DIY/low-cost designs) or high (commercial MFC systems)
Modularity/openness	limited	high (especially with open-source platforms)

**Table 2 sensors-25-07297-t002:** Summary of cumulative biogas production and daily production rates for three parallel substrate reactors (R_1_–R_3_). The first 24 h show the expected biological variability of batch anaerobic digestion, whereas subsequent intervals converge to zero as reactors reach the metabolic plateau. The final cumulative production is reported including *mean* ± *SD* and the corresponding 95% confidence interval.

Metric	*V_R_* _1_	*V_R_* _2_	*V_R_* _3_	*Mean* ± *SD*	95% CI (Final Only)
Final cumulative production (mL)	158.494	169.122	141.185	156.27 ± 11.65	127.4–185.2
Daily production 0–24 h (mL/day)	157.382	167.382	140.477	155.08 ± 13.67	—
Daily production 24–48 h (mL/day)	1.112	1.740	0.708	1.19 ± 0.52	—
Daily production 48–72 h (mL/day)	0	0	0	0 ± 0	—
Daily production 72–90 h (mL/day)	0	0	0	0 ± 0	—

**Table 3 sensors-25-07297-t003:** Final cumulative volumes after 90 h, blank-corrected; replicates R_1_–R_3_ are presented individually as well as by their mean value.

Measured Reactor Volume	Manual Method After 90 h (mL)	Automated Method After 90 h (mL)
*V_R_* _1_	153.0	157.53
*V_R_* _2_	148.0	168.16
*V_R_* _3_	139.5	140.23
Mean (*V_R_*_1_–*V_R_*_3_)	146.83	155.31

**Table 4 sensors-25-07297-t004:** Individual sources of measurement uncertainty balance for the calculation of the experimentally measured biogas production volume.

Source of Uncertainty	Symbol of Uncertainty	Uncertainty Value	$\mathrm{Calculation}\;\mathrm{for}\;V_{cum}=100\;mL$	Absolute Uncertainty (mL)
Repeatability	*u* _A_	0.1% FS of the sensor	$0.12\;mbar\;\times \;0.514\frac{mL}{mbar}\approx 0.0617\;mL$	---
Time granularity of the measurement process *	*u* _B1_	0.8%	$u_{B1}=100\;mL\;$ *×* 0.008 $u_{B1}=0.8\;mL$	exponential
Integration of initial dynamics *	*u* _B2_	0.5%	$u_{B2}=100\;mL\;$ *×* 0.005 $u_{B2}=0.5\;mL$	exponential
Evaluation method	*u* _B3_	1.0%	$u_{B3}=100\;mL\;$ *×* 0.010 $u_{B3}=1.0\;mL$	normal
Physical conditions **	*u* _B4_	0.5%	$u_{B4}=100\;mL\;$ *×* 0.005 $u_{B4}=0.5\;mL$	normal
Other influences	*u* _B5_	0.5%	$u_{B5}=100\;mL\;$ *×* 0.005 $u_{B5}=0.5\;mL$	normal
*u* _c_	1.547 mL			
*U* (*k* = 2)	U=2 ⋅1.547 mL≈3.09 mL			

* estimated value, ** estimated based on experimental measurements.

**Table 5 sensors-25-07297-t005:** Comparison of metrological and operational parameters of representative BMP measurement systems, including flow cell volumetric analyzers (AMPTS III), manometric pressure-based devices (VELP Maxi; Pérez-Vidal et al., 2025 [7]), semi-automated volumetric setups (AnaeroTech BMP/Nautilus), and the proposed hydrostatic platform. The table highlights key differences in resolution, uncertainty, drift, temporal resolution, sensitivity to CO_2_ solubility, and overall automation level.

Parameter	AMPTS III (Flow Cell)	VELP Maxi (Manometric)	AnaeroTech BMP/Nautilus (Volumetric)	Pérez-Vidal 2025 [7] (IoT Manometric)	Proposed Hydrostatic Platform
Measurement mode	Optical pulse-based flow cells	Headspace pressure → volume	Water-displacement burette	Continuous headspace pressure sensing	Differential pressure → hydrostatic column
Resolution	~0.05–0.1 mL	0.1–0.5 mL	0.5–2 mL	1 mbar sensor resolution (≈0.2–1 mL depending on headspace)	~0.06 mL
Short-term variability (*σ*)	0.05–0.10 mL	0.10–0.50 mL	0.50–2.00 mL	~1 mbar (noise)	0.06 mL
Expanded uncertainty (≈100 mL)	2–3%	5–10%	5–20%	7.6 mbar	~3.1%
Long-term drift	Very low	High (temp + solubility)	Operator-driven/irregular	High drift due to CO_2_ dissolution and temperature	<0.15%/24 h
Sensitivity to CO_2_ solubility	Low	Very high	Medium	Very high (headspace thermodynamics)	Eliminated (open hydrostatic loop)
Disturbance/gas loss	None	Moderate	Moderate	50.7 ± 12.9 mbar loss per reading	None
Temporal resolution	~1–2 Hz	10–60 s	Manual	1 × per hour	11 Hz internal/1 Hz logging
Automation	Full (industrial)	Partial	Low	Medium	Full (refill, venting, cloud)
Scalability	6–18 reactors	6	6–15	4	4 + modular
Typical cost	€20 k–35 k	€10 k–15 k	€4 k–8 k	€150–300	~€2 k
Primary limitation	Proprietary/high cost	Drift + solubility error	Manual variability	Strong dependence on headspace physics	No gas composition sensor (optional)

## Data Availability

The data generated and analyzed during the current study are available in the Zenodo repository at https://doi.org/10.5281/zenodo.17674470.

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
