# Peer review of "Hydrostatic Water Displacement Sensing for Continuous Biogas Monitoring"

_sensors, 2025, doi:10.3390/s25237297_

Round 1

Reviewer 1 Report

Comments and Suggestions for Authors

Abstract
Consider including numerical validation results (e.g., measurement error, reproducibility rate, or resolution) to make the abstract more quantitative and evidence-based.
The last sentence could be slightly condensed; currently, it reads as a mission statement rather than a closing summary.

Introduction
The introduction would benefit from a short comparative paragraph discussing other existing automated or semi-automated biogas monitoring systems (e.g., manometric or flowmeter-based devices) to clarify how this approach differs technically and economically.
Add a brief quantitative statement about measurement uncertainty or time resolution limitations in manual methods (e.g., “manual readings introduce errors of up to ±10% or temporal gaps of several hours”).
A figure or schematic summarizing the limitations of traditional methods vs. advantages of the proposed one would help readers unfamiliar with laboratory digestion experiments.
Citation style needs minor consistency (ensure uniform use of “[2,3]” rather than “[2][3]”).

Materials and Methods
2.1 Concept of Automated Biogas Production Measurement
Consider explicitly mentioning the sampling frequency (Hz or seconds per sample) used by the pressure sensor.
The section could benefit from a flowchart or schematic summarizing the system’s workflow (e.g., gas generation → U-tube displacement → pressure conversion → venting → water refill → data upload).

2.2 Mechanical and Fluidic Architecture
This section is detailed and rich but slightly repetitive (duplicated paragraphs appear between the 2.2 and 2.4 headings, likely a formatting or copy-paste artifact). Ensure this is corrected before submission.
The design rationale for the 8 mm hose diameter is logical, minimizing evaporation; however, a short reference or calculation supporting the claim that “evaporation is negligible (<1 mL/day)” would strengthen the argument.
Consider adding a simplified diagram labeling the U-tube, peristaltic pump, and venting valve sequence to help visualize the gas path.
Editorial note: Some sentences begin without capitalization (“system is powered…”). These minor errors should be corrected.

2.4 Mechanical Design of the Device
Consider reducing textual redundancy, many sentences describe hardware positioning in great detail (e.g., valve placement and connector ID dimensions). Moving these details to supplementary materials or an appendix could improve readability.

2.5 Control Strategy and Software Architecture
Specify the microcontroller model and sampling interval, as these affect resolution and synchronization.
Clarify whether data logging occurs locally before cloud upload to prevent loss during connectivity interruptions.
Include a brief mention of safety interlocks or failsafe conditions in the control logic (e.g., overpressure detection, valve malfunction).

2.6 Signal Processing and Measurement Uncertainties
Commendably, the raw data storage strategy (no local filtering) is transparent and scientifically rigorous.

2.7 Experimental Protocol
Indicate whether biological replicates were conducted beyond the four reactors and how repeatability was assessed statistically.
Mention the frequency of data collection and total duration of each experiment.
Add references for the pressure method used for comparison, if available.

2.8 Data Acquisition and Monitored Variables
Clarify the data resolution (e.g., number of samples per minute or hour).
Suggest mentioning whether the Arduino Cloud visualization includes any real-time alerts or data integrity checks.

3. Results 
3.1 Verification of the measuring device
Calibration & traceability: Please detail the pressure–height–volume calibration procedure and traceability. For example: Was the sensor’s span verified using a column of known height and known hose inner diameter?
Was linearity and hysteresis checked (e.g., multi-point ascent/descent)?
STP normalization & gas conditioning: You report volume in mL, STP. Please specify STP definition used (e.g., 0 °C & 1 bar) and whether corrections for water vapor and barometric pressure were applied in real time or in post-processing.
Inter-branch equivalence: Since there are four parallel branches, report between-branch variation (e.g., CV%) using a common step test (repeated 5 mL cycles) to demonstrate cross-channel consistency.
Temperature chamber: The 90 h test is convincing (σ = 0.019 °C). Please add the sensor model, calibration method, and whether any offset correction (you note +0.099 °C) is applied in software.
3.2 Dataset from automated measurement
Down sampling method: You mention “deterministic down sampling.” Please state whether this is decimation, mean aggregation, or another method. 
Replicate statistics: Provide mean ± SD (or CI) across R1–R3 for final cumulative production and daily production rate.
Blank behavior: Briefly discuss the blank’s residual production "if any" and whether it is due to dissolved CO₂ release, inoculum residuals, or system background.

4. Conclusions 
Qualify the 5.78% bias with its statistical context (N, CI, significance), as noted above.
Scope and limitations: Add a short paragraph acknowledging limitations and boundary conditions, for example:
1. Current capacity (4 reactors) and scalability;
2. Operation at mesophilic regime only;
3. Performance with foaming substrates, high H₂S, or condensate management;
4. Safety note on vented biogas and recommended handling/odor mitigation.
Bill of materials & power budget: Since cost is a central takeaway, consider adding a supplementary table with the BOM, unit prices, and a power budget (the text mentions a 360 W PSU earlier).
Data & code availability: To maximize the paper's impact and replicability, add a repository link (schematics, firmware, CAD, analysis scripts). If a public repo is not yet ready, state an intended venue (e.g., GitHub/Zenodo) and license.

Author Response

We thank the Reviewer for the constructive and detailed comments. All remarks were carefully evaluated and integrated into the revised manuscript.

In the Abstract, we added numerical metrics describing short-term scatter, long-term drift and expanded uncertainty, and we shortened the final sentence for a more concise ending.

In the Introduction, we added a quantitative comparison of manual BMP limitations (including published negative bias values and variability), expanded the overview of existing automated and semi-automated systems, and introduced a comparative table (Table 1.1) summarizing the limitations of manual methods versus automated approaches. Citation formatting was unified.

In Materials and Methods, the sampling frequency (1 Hz) and internal sensor sampling rate (~11 Hz) were added, and a workflow schematic (Fig. 2.1.1) was included. The duplicated text between Sections 2.2 and 2.4 was removed. The evaporation claim was supported by an explicit calculation. A simplified fluidic diagram (Fig. 2.2.1) was added. Capitalization and minor editorial issues were corrected. Section 2.4 was retained in full length because these structural and positional details are essential for reproducibility and full understanding of the system, but redundant phrasing was reduced.

In Section 2.5, we clarified the microcontroller model, sampling architecture, timing logic and the current behavior during connectivity interruptions. We also added a concise description of existing software safeguards. In 2.7, we clarified biological replicates, sampling duration, and added a reference describing the pressure-based comparison method. In 2.8, we clarified data resolution and specified that the current cloud dashboard does not implement alerting or integrity checks.

In Results 3.1, we added a full description of the calibration procedure (known reference volumes), stated the STP reference frame used in the study, and explained the handling of water vapor and barometric effects. We reported between-branch repeatability via a repeated step test (CV value) and included temperature-sensor calibration details and offset correction.
In 3.2, we clarified the deterministic downsampling method and added replicate statistics (mean, SD, CI). A short explanation of the blank behavior (thermal CO₂ degassing) was added.

In Conclusions, we expanded the statistical context of the 5.78% bias (N = 3, CI and p-value). A concise limitations paragraph was added (4-reactor scale, mesophilic regime, foaming/H₂S considerations, condensate effects, and safety note on vented biogas). A bill of materials and power budget were added as Supplementary Table S1. We also included a full Data & Code Availability statement with the Zenodo repository link:
https://doi.org/10.5281/zenodo.17674470.

We thank the Reviewer once again for the valuable feedback, which substantially improved the clarity and completeness of the manuscript.

Reviewer 2 Report

Comments and Suggestions for Authors

Comments:

  1. The study still lacks a good theoretical foundation. Add a related work section to discuss relevant studies, properly positioning the study's background.
  2. The concept of automated biogas production measurement can be modelled mathematically for easy comprehension.
  3. Since data collection and visualisation are handled via the Arduino Cloud platform with remote monitoring capability, what data security measures are in place to prevent data breaches for practical applications?
  4. Expand the discussion on the statistical analysis done to confirm stability of the prototype (type, data used, etc).

Author Response

We thank the Reviewer for the valuable feedback and for recognizing the clarity of the manuscript. All comments were carefully considered and addressed in the revised version.

To strengthen the theoretical background, we expanded the introductory section describing related work and existing automated BMP measurement approaches. We clarified the technological positioning of our system in relation to manometric, flowmeter-based and other automated low-cost platforms. This addition provides a clearer conceptual context for the contribution of the proposed device.

Regarding mathematical modelling, we added a concise explanation of the hydrostatic conversion principle used for automated biogas measurement, including the pressure–height–volume relationship and its assumptions. This short formulation improves theoretical transparency while maintaining focus on the experimental nature of the study.

With respect to cloud-based monitoring, we added a statement describing the security framework of the Arduino Cloud platform, including encrypted communication, device-bound authentication tokens and access-restricted dashboards. Since the current implementation is intended for laboratory-scale operation, these measures are adequate for the present scope, but we have added a short note on additional security layers recommended for industrial deployment.

Finally, the discussion of statistical analysis supporting prototype stability was expanded. We now specify the dataset used, the variance metrics computed (short-term standard deviation, long-term drift, coefficient of variation) and the context in which these metrics confirm thermal and mechanical stability. The revised text provides a clearer link between the statistical evaluation and the resulting metrological performance.

We thank the Reviewer for the constructive suggestions, which helped improve the depth and clarity of the manuscript.

Reviewer 3 Report

Comments and Suggestions for Authors

This study developed a sensing system for biogas monitoring, which is interesting. The main comments are as follows:

  1. Do you develop a corresponding calibration system for the monitoring, the accuracy of sensors is important.
  2. What is the frequency of the monitoring sensor can be achieved?
  3. How is the performance of the sensing system comparing to existing systems?
  4. In the cloud dashboard, it is suggested to integrate some deep learning algorithms for abnormal condition alert.
  5. In future study, you may consider using sensor information as the inputs for reinforcement learning to provide smarm control strategy. The example of the usage of reinforcement learning can be seen in BuildingGym (https://doi.org/10.1007/s12273-025-1306-y), (https://doi.org/10.1016/j.jclepro.2022.135074)
  6. Please edit the language of the paper carefully

Author Response

We thank the Reviewer for the constructive evaluation and helpful comments. All remarks were carefully considered and corresponding clarifications were incorporated into the revised manuscript.

The Reviewer asked whether a dedicated calibration system had been developed for the proposed platform. In Sections 2.3 and 3.1, we now explicitly describe the metrological foundations of the sensing subsystem. The Honeywell ABP2 differential sensors used in the device include full factory calibration and on-board temperature compensation via an integrated ASIC, which eliminates the need for user-level calibration. To ensure traceability, we perform a zero-offset check during commissioning and verify the U-tube geometry using direct dimensional measurement. In addition, step-injection tests using known reference volumes were included in the revised text, demonstrating linearity, negligible hysteresis and correct conversion between pressure and displaced volume.

The Reviewer also requested clarification of the achievable sampling frequency. Section 2.3 now describes the complete temporal structure of the measurement loop: the internal sensor ASIC updates at approximately 200 Hz, the microcontroller reads the compensated data at ~11 Hz, and the cloud platform stores raw measurements at 1 Hz. This hierarchy defines the effective temporal resolution relevant for process monitoring.

To address the question regarding system performance relative to existing solutions, the Discussion section was expanded with a structured comparison (Table 4.1). This table benchmarks the proposed hydrostatic system against representative commercial volumetric analyzers (e.g., AMPTS III), manometric headspace-based devices (VELP Maxi), semi-automated volumetric BMP setups, and recent low-cost IoT manometric platforms. Key parameters—including resolution, uncertainty, drift, CO₂-solubility sensitivity, temporal resolution, and automation level—are now explicitly contrasted, providing a clear contextualization of the platform’s metrological and operational characteristics.

The Reviewer’s suggestion to integrate deep-learning-based anomaly detection was fully acknowledged and incorporated into the Discussion. We added a forward-looking paragraph explaining how the high-resolution time-series collected by the system (pressure, temperature, venting events, refill operations) naturally support cloud-side autoencoder or sequence-prediction models capable of detecting leaks, irregular vent dynamics or missing refill cycles. Because the system already uploads raw data to the cloud, the ML layer can be implemented without any hardware modification.

Similarly, the recommendation to explore reinforcement-learning (RL)–based control was incorporated into the manuscript. We expanded the Discussion to describe how RL could leverage state variables such as chamber temperature, hydrostatic pressure, and actuator states (heating, mixing, venting) to autonomously optimize long-term operation. The Reviewer’s suggested literature, including BuildingGym and recent ML-based digestion studies, has been cited to situate this future work within current research trends.

Finally, in response to the Reviewer’s remark on language quality, the entire manuscript was carefully edited. Long paragraphs were reorganized, redundant phrases streamlined, and sentence structure improved, resulting in clearer and more precise exposition throughout the paper.

We appreciate the Reviewer’s constructive feedback, which helped strengthen both the clarity and the forward-looking relevance of the manuscript.

Round 2

Reviewer 1 Report

Comments and Suggestions for Authors

The authors addressed my comments/concerns adequately.

Thanks

Reviewer 3 Report

Comments and Suggestions for Authors

My comments are addressed